# Ethnic diversity fosters the social integration of refugee students

Zsófia Boda ●[1,8] ✉, Georg Lorenz ●[2,3,7,8] ✉, Malte Jansen[2,4], Petra Stanat[2,5] & Aileen Edele ●[5,6]

Forced migration has become a global megatrend, and many refugees are school aged. As social integration is key to their wellbeing and success, it is pivotal to determine factors that promote the social integration of refugee youth within schools. Here, using a large, nationally representative social network dataset from Germany, we examine the relationships of refugee adolescents with their peers (304 classrooms, 6,390 adolescents and 487 refugees). We find that refugee adolescents have fewer friends and are more often rejected as desk mates than their classmates. Crucially, however, they are less rejected in more diverse classrooms. This results from two basic processes: (1) more opportunities to meet other ethnic minority peers, who are more accepting of refugees in general and (2) higher acceptance of refugee adolescents by ethnic majority peers in more diverse settings. Our results can help promote the social adjustment of young refugees in school and mitigate the negative consequences of prejudice.

While migration has always been part of human history, the proportion of displaced people has grown rapidly in the last decade, and now exceeds 1% of the world's population[1]. Due to political instability, armed conflicts, persecution, demographic change, economic deprivation and severe effects of climate change, refugee migration is expected to remain a megatrend. A large proportion of refugee migrants in Europe are children and adolescents under the age of 18 years (ref. 2). These young people need access not only to formal education, but also require positive peer relationships because these are essential determinants of their adaptation[3]. Yet we know very little about the peer relations of refugee migrants. This is a major research gap, given that peers are highly important socialization agents in adolescence, influencing young people's lives in many ways[4–6]. Moreover, the acculturation of refugees underlies special conditions due to, for instance, mental stress, insecure legal status and interrupted educational careers[7].

The social integration of adolescents refers to positive and supportive relationships, as indicated by friendships and a lack of peer rejection[8,9]. Social integration improves adolescents' wellbeing[10],

health[11] and educational achievement[12]. In contrast, low levels of social integration can have severe consequences for adolescents' psychological wellbeing[13] and physical health[14].

Friends are pivotal to social integration as they provide social capital, which includes resources such as valuable information and social support[15,16]. However, the resources embedded in co-ethnic friendships often differ from those embedded in inter-ethnic friendships[17]. In particular, immigrant students benefit from social contact with majority group members[18]. Such contacts provide access to resources such as exposure to the host country's language. Consequently, they enhance immigrant adolescents' opportunities to acquire critical resources such as language skills[19] and, ultimately, success in the education system[20] and labour market[21]. Therefore, in addition to having positive peer relationships in general, establishing relationships across ethnic boundaries constitutes another key component of social integration for minority students[22].

For refugee adolescents, language difficulties and consequences of traumatic experiences can provide barriers to peer acceptance[23,24].

[1]Department of Sociology and Institute for Social and Economic Research, University of Essex, Colchester, UK. [2]Institute for Educational Quality Improvement (IQB) at Humboldt-Universität zu Berlin, Berlin, Germany. [3]Department of Sociology, Leipzig University, Leipzig, Germany. [4]Centre for International Student Assessment (ZIB), Munich, Germany. [5]Berlin Institute for Integration and Migration Research (BIM), Berlin, Germany. [6]Department of Education Studies, Humboldt-Universität zu Berlin, Berlin, Germany. [7]Present address: University of Potsdam, Potsdam, Brandenburg, Germany. [8]These authors contributed equally to this work: Zsófia Boda and Georg Lorenz. ✉e-mail: zsofia.boda@essex.ac.uk; georg.lorenz@uni-potsdam.de

Moreover, social integration is not a one-sided process. Instead, it depends on multiple actors' simultaneous attitudes and behaviours[25], with the attitudes and behaviours of peers being crucial. As these are affected by the social environment, it is pivotal to consider the role of the school and classroom context to better understand the social integration of refugee adolescents.

One contextual aspect that is likely to be particularly important for social integration is school ethnic diversity. In more diverse school settings, refugee students have more opportunities to interact with peers who also have an immigrant background. These ethnic minority peers, in comparison to peers from the ethnic majority group, tend to have less inter-group anxiety[26]. Additionally, ethnic minority peers might have more positive attitudes towards refugees than majority group members because they are more likely to share experiences of being perceived as culturally distinctive (for example, due to being Muslims) and to face similar challenges in achieving their educational goals (for example, due to language barriers)[27]. Thus, a higher level of ethnic diversity at school might facilitate refugee students' social integration because non-refugee ethnic minority members should be more likely to associate with them than majority group members.

Additionally, school ethnic diversity might improve the social integration of refugee students by promoting preferences for inter-ethnic relationships among the majority group. Negative attitudes and prejudice towards immigrants (including refugees) are widespread in Western societies[28]. However, a well-established finding from social psychology is that inter-group contact increases familiarity with out-group members[29,30] and reduces prejudice as well as racial and ethnic intolerance[31,32]. Inter-group contact thus leads to more positive inter-group attitudes[33], particularly among members of the majority group[34]. As a result, majority group members form increasing numbers of inter-ethnic friendship ties when ethnic diversity is higher[35] (though this increase is not usually proportionate to increased opportunities[36,37]), and aggression towards ethnic minority members decreases[38]. These processes might also benefit the social integration of refugee students. Perceived ethnic threat—the perception by majority group members that a minority group threatens their dominant position within communities, which can cause them to reject minority group members—could counteract the positive effects of ethnic diversity on inter-ethnic relationships. However, ethnic threat mostly arises when a minority group is relatively large[36,39], which is not the case for refugee students attending schools in Western destination countries, including Germany[40].

In this Article, using techniques for social network analysis and focusing on Germany, where the number of people who filed for protection rose by 2 million between 2014 and 2021 (ref. [41]), we examine the social integration of refugee students among their classmates. Our analyses are based on the largest dataset on refugee students' social networks currently available. The full data include complete friendship and desk mate rejection networks of 39,154 secondary school students in 1,807 school classes in Germany, with 342,114 friendships and 161,430 rejection relations measured. In the data, we identified 487 refugee adolescents in 304 classrooms, including 6,390 students in total. We examine how often refugee students are named as a friend or rejected as a desk mate and by whom. Moreover, we determine how different levels of ethnic diversity in the classroom affect friendship and desk mate rejection patterns.

Our identification strategy leverages the involuntary choice of residence among refugees in Germany. Upon arrival, refugees are allocated to German federal states on the basis of governmental quotas that consider the states' tax revenues, population size and accommodation capacities[42]. Consequently, the allocation of refugees is related to the local population size but largely independent of refugees' characteristics and the number of immigrants in a municipality[43]. Until 3 years after granting the status of recognized refugees, refugees are obliged to stay in the municipality (federal states of Baden Wurttemberg, Bavaria,

Berlin, Bremen, Hamburg, Hessen, North Rhein-Westphalia, Saarland, Saxony and Saxony Anhalt) or in the federal state (federal states of Brandenburg, Lower Saxony, Mecklenburg-Western Pomerania, Rhineland-Palatinate, Schleswig-Holstein and Thuringia) to which they had been assigned upon arrival. The restricted freedom of movement is only lifted if a person or a close relative starts a regular job, enrolls in a university or starts vocational training. However, these criteria are very rarely met. Between 2016 and 2018, for instance, only 8% of the refugees in Germany moved to another state[43]. Due to these allocation procedures, refugees' residence is largely independent of their personal characteristics and from key context characteristics such as ethnic diversity. This allows us to gain general insights into how variation in the receiving context affects the social integration of newly arrived immigrants.

In the following, we first describe refugees' social integration within schools and examine whether their friendship and desk mate rejection networks differ from those of majority students and other (non-refugee) ethnic minority students. Then, we investigate why refugee adolescents who attend ethnically diverse schools are socially better integrated than their counterparts in less diverse schools.

## Results

### Friendship and rejection of refugee students

Descriptive analyses of social ties show that refugee students are less socially integrated than their ethnic minority and majority peers. Figure 1a,b shows friendship and desk mate rejection nominations towards refugee students, as reported by their classmates with varying immigrant status. We distinguish among refugee classmates, native classmates (student and both parents born in Germany; that is, the ethnic majority) and two groups of non-refugee ethnic minority classmates: first-generation immigrants (born abroad) and second-generation immigrants (born in Germany, with at least one parent born abroad). The proportions of classmates who chose a student as a friend or rejected a student as a desk mate are shown on the horizontal axis; the density curves indicate the distribution of friendships and rejections of students by immigrant status. We find that refugee students have fewer friends and are more often rejected as desk mates than their peers. While non-refugee students are chosen, on average, by 45% of their classmates as friends, refugee students are chosen by only 33%. Moreover, non-refugee students face an average rejection rate of 21% from their classmates, while refugee students are rejected as desk mates by 31% of their classmates. Friendship and rejection rates, in contrast, show little variation within the non-refugee groups (42–45% for friendship and 20–23% for rejection). The differences between refugees and non-refugees are statistically significant for both indicators of social integration (friendship: $P < 0.001$, $t = 12.41$; rejection: $P < 0.001$, $t = -10.81$; the differences between the refugee group and each non-refugee group are also statistically significant). The results hold when we control for gender, age, academic achievement, language skills and length of stay in Germany. In these multivariate models, members of all non-refugee groups except first-generation immigrants show significantly higher levels of social integration than refugees (friendship nominations among second-generation immigrants: $P < 0.001$; friendship nominations among natives: $P < 0.001$; rejection nominations among second-generation immigrants: $P < 0.001$; rejection nominations among natives: $P < 0.001$) (Supplementary Table 1 in Supplementary Appendix A provides the full results).

### Refugee students' social networks

We continue by examining how refugees' social integration varies with classroom ethnic diversity. We do this by looking at ego networks of refugees. That is, we calculate how many peers nominate refugee students as friends and how many peers reject them as desk mates in classrooms with low, medium and high ethnic diversity. These diversity levels are represented by the lowest, medium and highest third of the

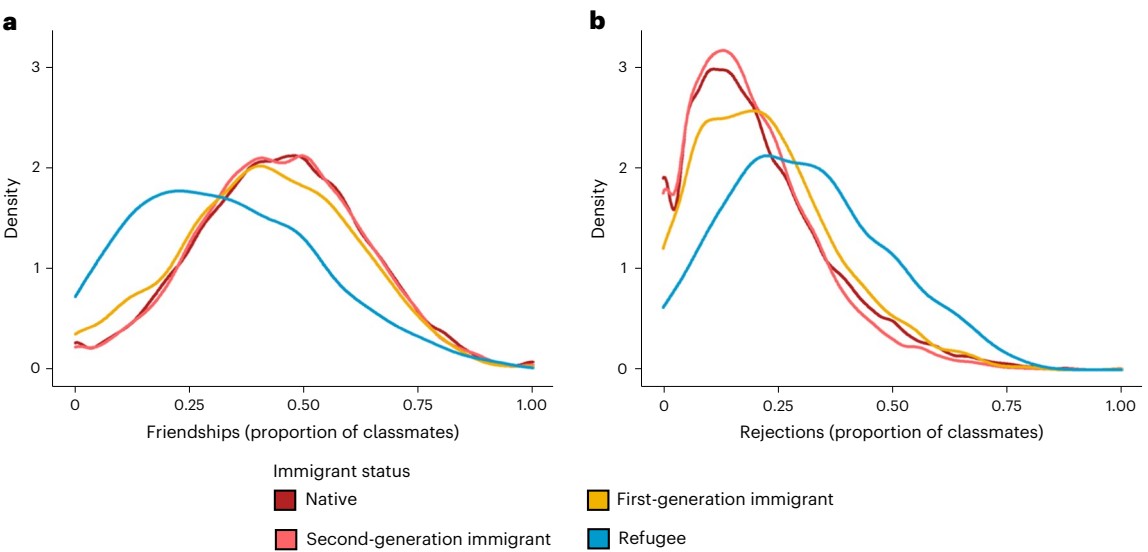

**Fig. 1 | Friendships and rejections among refugee adolescents based on their peers' immigrant status. a,b,** Density plots showing friendship (**a**) and desk mate rejection (**b**) nominations towards refugee adolescents received by native, first-generation immigrant, second-generation immigrant and refugee classmates. $N_{students} = 39,154$; $N_{classrooms} = 1,807$.

diversity distribution in the analysed sample, respectively. The categories define different diversity levels relative to each other and are not meant to be understood as absolute measures. Ethnic diversity is defined as the proportion of pairs of students who have different countries of origin[36,37,44]. Emanating from the assumption that non-refugee ethnic minority students may be more likely to develop positive social relations with refugees than native students, we also analyse the ethnic composition of refugee students' social networks in these three types of classrooms. Figure 2 visualizes typical friendship networks (Fig. 2a–c) and desk mate rejection networks (Fig. 2d–f) of refugee students in three classroom types (low, medium and high classroom diversity). The bar charts indicate how many classmates from each immigration status group name refugee students as friends or reject them on average in each diversity setting (low, medium and high). The network plots were constructed on the basis of the (rounded) frequencies shown in these bar charts.

We find that refugee students receive a similar average number of friendship nominations across the three diversity settings: 5.12 nominations in low-diversity settings (Fig. 2a), 5.59 in medium-diversity settings (Fig. 2b) and 6.00 in high-diversity settings (Fig. 2c). The difference is significant only between the low- and high-diversity settings ($P = 0.04$, $t = −2.04$). However, the composition of friendship networks co-varies substantially with diversity. As diversity increases, refugee students' networks include more ethnic minority peers and fewer native peers. This is not surprising, given that classes with higher ethnic diversity include more ethnic minority students by definition. Importantly, Fig. 2 also shows that a refugee student has, on average, one refugee friend across all diversity settings. This holds even though the majority of the classrooms include only one or two refugee students, with a mean of 1.6 refugee students per class in the analysed sample. Thus, refugee students appear to be very likely to befriend each other if more than one of them is present in a classroom. This finding reflects the well-established phenomenon of homophily[45], which describes that people tend to build social ties with those who are similar in terms of salient attributes, such as ethnic origin or flight experience.

Next, we turn to associations between-classroom diversity and the desk mate rejection networks of refugees. Refugee students are rejected less often as desk mates in more diverse classrooms. In low-diversity settings (Fig. 2d), refugee students are rejected as desk mates by 7.48 of their classmates on average, of whom 6.79 are native.

In medium-diversity settings (Fig. 2e), the average number of rejections is reduced to 5.94 peers, of whom 4.19 are native (Fig. 2b). In high-diversity settings, refugee students face rejection as desk mates by only 4.32 classmates, of whom 1.64 are native (Fig. 2c). Overall, refugee students are rejected as desk mates by 42% fewer classmates in high-diversity settings than in low-diversity settings, with the difference in rejection rates being significant between each pair of diversity settings (low versus medium diversity: $P < 0.001$, $t = 3.46$; medium versus high diversity: $P < 0.001$, $t = 3.97$; low versus high diversity: $P < 0.001$, $t = 7.26$).

**Preferences for befriending and rejecting refugee students**

Figure 2 shows students' social networks in classrooms with varying diversity. However, it does not provide any indications about the underlying processes. Are refugees in higher-diversity settings better socially integrated simply because of the presence of more ethnic minority students who may be more likely to accept them or because they develop more positive relations with peers from all ethnic groups, including native students? To answer these questions, we apply linear regression models specifically developed to analyse social network data[46]. These models consider that observations in social networks (that is, social ties) are not independent of each other. To compare all types of nominations, we add a variable capturing every possible nomination type on the basis of the combination of the sender's and the receiver's immigrant status. We control for the gender, age, length of time living in Germany, academic achievement and language skills of both the tie sender and the tie receiver, as well as their match with regard to these variables and their country of origin.

Figure 3 shows the subset of the findings for refugee students' social ties (the full results are provided in Supplementary Table 2 in Supplementary Appendix A). The figure displays the estimated probabilities of a refugee student being chosen as a friend (Fig. 3a) and being rejected as a desk mate (Fig. 3b) by classmates from each immigrant status category (as indicated by the colour of the line) and depending on the level of classroom diversity (as shown on the horizontal axis). The black line represents the baseline nomination tendency (that is, the probability of native → native nominations) in different diversity settings. We highlight that ethnic diversity is included as a continuous variable in these models and not as a categorical variable (as presented in Fig. 2). The histogram above each plot shows its distribution across the classrooms.

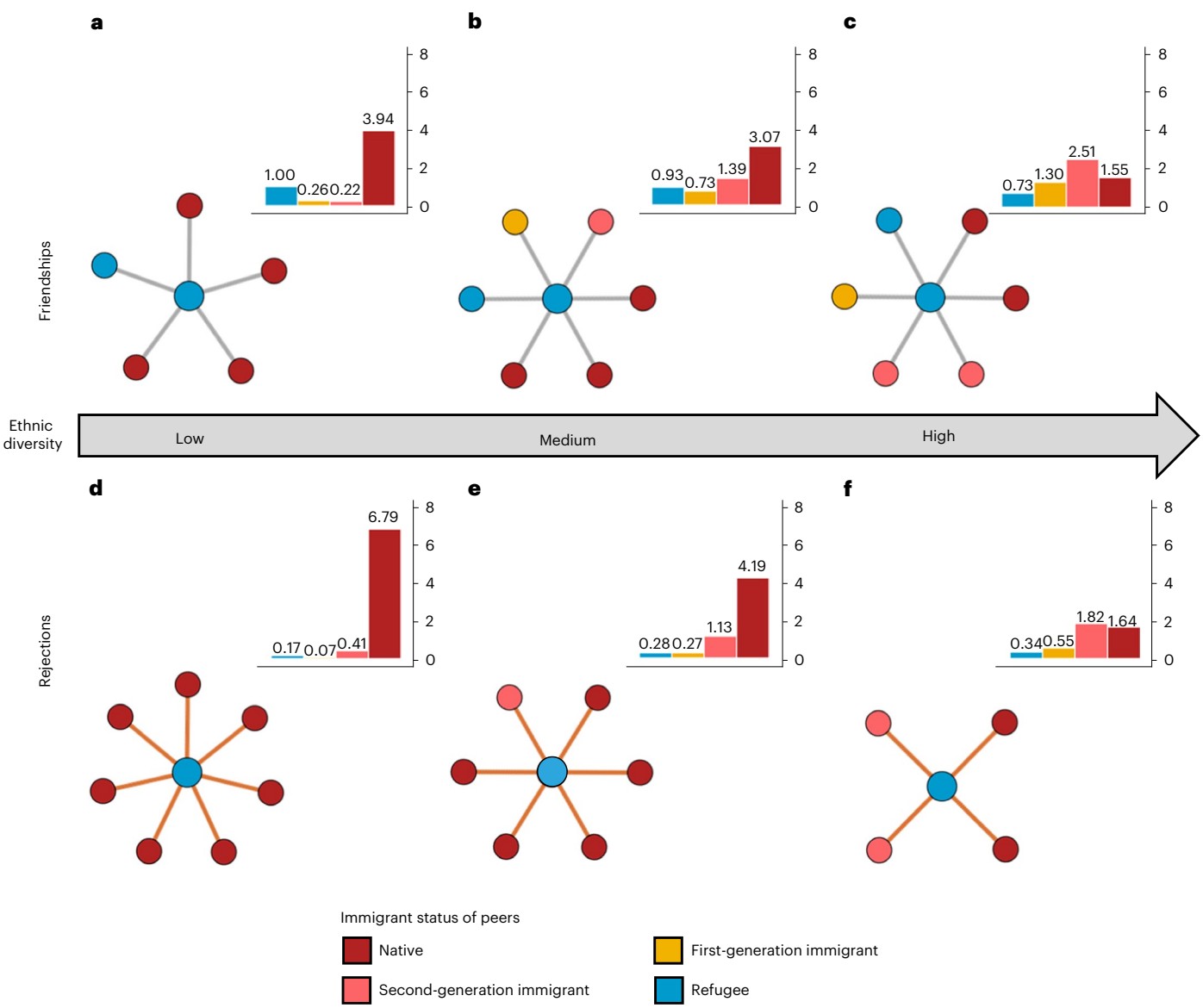

**Fig. 2 | Typical friendship and rejection ego networks of refugee students.**
**a**–**f**, Bar plots showing the rounded average number of friendship (**a**–**c**) and desk mate rejection (**d**–**f**) nominations of refugee students by each immigrant status group. The network plots map the average number of nominations (for example, the total number of friends). In addition, the network plots are a proportionate representation of the ethnic composition of nominations of refugee students (for example, the number of native friends). Note that not every bar can be represented in the network plots by the value of the closest integer to its actual mean due to our primary goal to represent the total number of nominations accurately. $N_{students}$ = 5,328; $N_{classrooms}$ = 237.

Figure 3a reveals that, among non-refugee classmates, native students are the least likely to choose refugee students as friends in all diversity settings. Second-generation immigrant students take an intermediate position, while first-generation immigrant students are most likely to nominate refugee peers as friends, even more often than the baseline probability. Remarkably, in more diverse classrooms, students of all non-refugee backgrounds tend to name refugee students as friends more often than in less diverse classrooms, with native students showing the largest increase. The positive effect of diversity (shown by the slopes) is significant for native students ($P$ = 0.045). For refugee students, however, this effect is significant and negative ($P$ < 0.001), meaning that they nominate each other less often in more diverse settings. These findings indicate a substantial role of school diversity in refugees' friendships with native peers.

Figure 3b shows that, in low-diversity classrooms, native peers reject refugee students as desk mates with the highest probability, followed by second-generation immigrant, first-generation immigrant and refugee students. Rejections as desk mates from native and second-generation immigrant students become less likely with increasing diversity, whereas first-generation immigrant and refugee students reject their refugee peers more when diversity increases (as shown by the slopes). These relationships between diversity and the rejection of refugees are significant for all groups except second-generation immigrants (natives: $P$ = 0.006; first-generation immigrants: $P$ = 0.010; refugees: $P$ = 0.023). In classrooms characterized by the highest levels of ethnic diversity, the immigrant status groups differ only marginally in terms of their rejection rates of refugee students (8% maximum difference). This suggests that, in more diverse school settings, group differences in the tendency to reject refugee students level out. We provide additional information on the statistical significance of the effects shown in Fig. 3 in Supplementary Appendix F.

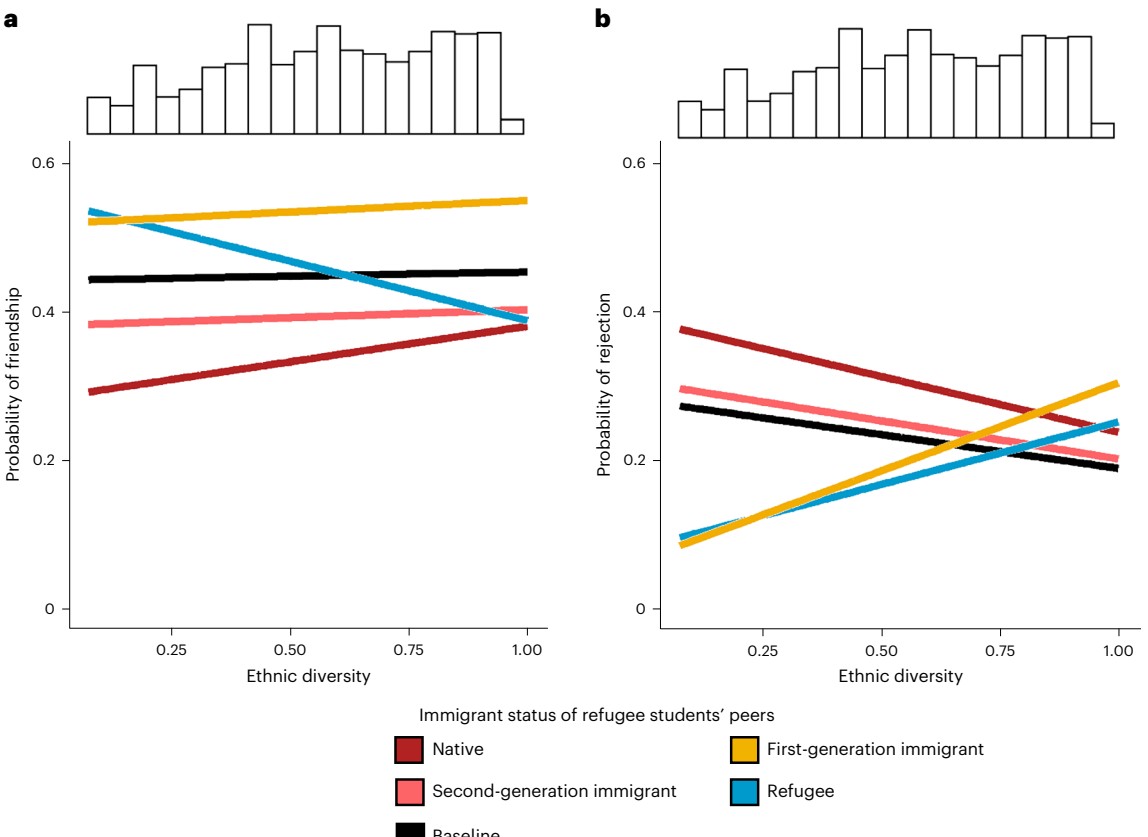

**Fig. 3 | Friendships and rejections among refugee students by classroom ethnic diversity. a,b,** Results from MRQAPs showing the probabilities of friendships with (**a**) and desk mate rejections of (**b**) refugee students (for details, see the 'Statistical analysis' section). The model controls for whether the tie sender and receiver have the same country of origin, the same gender, a similar age, a similar length of stay in Germany, similar language skills and similar academic achievement. Additionally, ego and alter effects of these control variables are controlled (for details, see 'Measurement'). The histograms above the plots represent the distribution of ethnic diversity across the classrooms. Supplementary Table 2 in Supplementary Appendix A presents the full results from the corresponding MRQAPs. $N_{students} = 6,390$; $N_{classrooms} = 304$.

## Discussion

Between 2014 and 2021, the number of people filing for protection in Germany increased by 2 million (ref. 41). Substantial proportions of refugees are children and adolescents: in 2021, 49% of the refugee applicants were younger than 18 years old[47]. This study demonstrates that a few years after their arrival, refugee adolescents are less socially integrated within their classrooms than their ethnic majority and non-refugee ethnic minority peers. This constitutes a serious risk factor for their educational success and psychosocial adjustment[13,48]. However, in more ethnically diverse classrooms, refugee adolescents are socially better integrated: they tend to have more friends and are rejected significantly less frequently as desk mates than in less diverse classrooms. This is revealed in descriptive analyses and multivariate social network models. The latter control for various factors relevant to social network dynamics, among them academic achievement and German language skills, which may make refugees less desirable as desk mates for their peers even in the absence of actual dislike. Notably, refugee students' higher social integration in more diverse classrooms is not solely due to the preferences and higher shares of ethnic minority students, but variation in the preferences of majority group adolescents for befriending and rejecting refugee peers across social contexts. In more diverse contexts, majority group adolescents reject refugees as desk mates less often and tend to nominate them as friends more often than in less diverse contexts.

The findings suggest that ethnic diversity promotes the integration of refugee students and that this is due to two basic mechanisms. First, non-refugee ethnic minority adolescents, particularly first- and second-generation immigrant students, show a higher baseline tendency to befriend refugee peers and a lower baseline tendency to reject them as desk mates than majority adolescents (independent of the school's ethnic diversity). Consequently, a higher share of ethnic minority students in more diverse classrooms supports refugee students' social integration. Second, majority group students build more positive relationships with refugee students when ethnic diversity increases (especially concerning decreased rejection rates). This might be due to lower levels of prejudice and more positive inter-group attitudes caused by increasing familiarity with outgroup members in more diverse settings[29,30] and/or the prevalence of social norms that promote inter-group contact in these settings[49].

The evidence that refugees are better integrated in classrooms with higher levels of diversity is in line with the inter-group contact theory, which posits that outgroup contact leads to more positive outgroup attitudes and relationships[33,50]. At the same time, our results suggest that processes of (perceived) ethnic threat[44,51] are negligible factors in refugee adolescents' social integration: ethnic majority students do not reject refugees more often in more diverse contexts. A possible explanation for the lack of ethnic threat is that, although the influx of refugees was high in recent years, the proportion of refugee students in German classrooms is typically low.

As newly arrived refugees cannot choose their place of residence and their mobility is highly restricted within 3 years after arrival in Germany, it can practically be ruled out that the relationship between social integration and classroom diversity is due to selection effects on the side of the refugees. Selection effects on the side of the classrooms,

however, are still possible (for example, teachers and peers may differ across classrooms with different levels of diversity). As a robustness check, we included classroom-level fixed effects in the model, which allows for ruling out general classroom-level processes to a certain extent. In another robustness model, we included federal–state-level diversity measures in addition to classroom-level measures. The results indicate that small-scale diversity differences at the school level are more pronounced than the effects of regional differences in ethnic diversity on refugee adolescents' social integration. Yet, regional variation in ethnic diversity is also significantly associated with fewer rejections of refugee students by ethnic majority peers (see the 'Robustness checks' subsection in the 'Statistical analysis' section), suggesting that school and regional context effects jointly affect refugee adolescents' social integration.

Another factor that might contribute to lower levels of social integration of refugees is that they have joined the classroom later than other students. Our multivariate models control for how long each student has been in Germany but not how long they have been in the classroom. Our data provide no information on when the refugee students had first entered their classes. Therefore, we cannot rule out that less time spent in the classroom (beyond less time spent in the country) partially explained the lower levels of social integration of refugees. However, this mechanism is unlikely to influence the role of diversity in this process.

Noteworthily, the diversity effects are more pronounced and robust for rejections than for friendship nominations towards refugees. This is in line with previous findings from social network studies. Endogenous processes, such as befriending friends of friends (transitivity), can increase ethnic segregation in friendship networks even when students from different ethnic groups do not dislike or reject each other[52,53]. Once friendships between co-ethnic peers exist, friends of friends will also be more likely to come from the same group. Inter-ethnic friendships, which are less likely to be embedded in friendship clusters, will more easily dissolve. Hence, even when peers tend to have more positive attitudes towards refugees in more diverse contexts, endogenous network processes may prevent increased levels of friendship integration. In contrast, more positive attitudes towards refugees in more diverse classrooms should translate into less rejections in more direct ways. Our results thus support earlier findings that stimulating friendships between ethnic minority students and their peers is complex and not an automatic result of reduced prejudice and inter-group rejections.

It should also be noted that, while ethnic diversity is generally beneficial for refugees' social integration, it is not positively associated with all types of inter-group relations. In particular, first-generation immigrant students are more likely to befriend but also to reject their refugee peers in more diverse classrooms. This may stem from a higher acceptance of outgroup members in such contexts paired with an attempt of students who also immigrated to Germany to distance themselves from their refugee peers. The finding suggests that beneficial and detrimental effects of contact can co-exist and highlights the importance of capturing different aspects of social relationships simultaneously. However, we only have a relatively low number of first-generation immigrant students in our sample ($N = 487$), and by definition, only a small proportion of them attend classrooms with low-diversity levels. Consequently, the findings related to diversity effects on first-generation immigrants' social-tie-creation behaviour should be treated with caution. Refugee students' decreasing likelihood of befriending other refugees and their increased likelihood of rejecting them in more diverse classrooms might also reflect an attempt to distinguish themselves from their ingroup. Alternatively, it might be a consequence of other students' greater openness towards refugees in more diverse classrooms and their extended opportunities to build friendships on the basis of other characteristics than immigration status. Overall, our findings suggest that a high level of diversity results in

a lower degree of overall rejection of refugees, yet, the pattern of who rejects and befriends them also changes.

From a political perspective, the finding that ethnically diverse school settings provide better conditions for the social integration of refugees challenges critical views of multi-culturalism. Our results imply that placing young refugees in school environments that are already ethnically diverse can, to some degree, promote their social adjustment due to social acceptance by other ethnic minority peers, but also due to a reduced rejection by majority group members, who are more accepting of refugee peers in more diverse classrooms. Given that positive contact with majority group peers is critical for immigrant students' academic success and school adaptation, assigning refugees to ethnically diverse schools and classrooms might also foster their economic integration and future life opportunities[18].

Deducing policy recommendations from the study's findings is particularly difficult for countries and regions with considerable variation in diversity levels. Taken at face value, the results suggest that it would be best for refugee students to attend highly diverse schools. However, policy advice needs to take various outcomes into account, including student academic achievement. Refugee students' language development, for instance, benefits from a high proportion of language majority students in the attended school classes, and it is a crucial determinant of their academic achievement[54]. Moreover, steering refugee students into more diverse schools would increase segregation, with some schools being attended by high proportions and others by only very few or no minority students. In such a scenario, inter-group contact would continue to be low in contexts marked by low diversity, and the processes allowing for more positive inter-group attitudes and higher social integration reported in this study would hardly occur there. To eventually blur ethnic boundaries, it would be important to ensure that diversity spreads out. Our findings clearly show that, in this process, special attention must be paid to refugee students in low-diversity settings. They seem particularly vulnerable to not being socially accepted in contexts with low levels of diversity, and institutional support for their social integration is vital. Therefore, school principals and teachers need to be aware of these challenges and support integration by, among other things, encouraging cooperation, setting up common goals and showing explicit support for mixing ethnic groups[33].

## Methods

### Ethical compliance

The data collection is part of the educational monitoring strategy ratified by the Standing Conference of the Ministers of Education and Cultural Affairs of the Länder in the Federal Republic of Germany. The German school laws and regulations state that participation in large-scale school assessment studies aiming to assure educational quality (including the Institute for Educational Quality Improvement (IQB) Trends in Student Achievement studies, but also the PISA, TIMSS and PIRLS studies) can be obligatory for schools, school principals, teachers and students. In accordance with these laws and regulations, the study participants are informed about the general content of the achievement tests and surveys in advance of this monitoring. The Ministries of Education of each federal state approve the data collection, including all of the instruments. This approval procedure considers ethical aspects as well as data protection requirements according to German law. For the 2018 Trends in Student Achievement study, the Ministries of Education agreed to waive consent requirements and endorse compulsory participation for the above reasons. However, as school laws differ between the 16 federal states, the exact procedure varied: while participation in the achievement test was mandatory in all states, participation in the questionnaires was mandatory in some states (although students were free to skip questions they did not want to answer) and voluntary in others.

## Data

Our analysis uses data from the Trends in Student Achievement study 2018 conducted by the IQB[55,56]. The study measured the academic achievement of ninth graders in Germany in mathematics and science and collected questionnaire data, including information on students' family background. The sample was selected by randomly drawing schools on the basis of the distribution of secondary school types (for example, academic track, intermediate track and comprehensive track) in each federal state of Germany. Subsequently, classrooms were randomly drawn in each school (one in academic track secondary schools and two in all other schools). Participation in the achievement tests was mandatory in all public schools, resulting in a participation rate of 92.4%. Completing the student questionnaire was voluntary in some federal states and mandatory in others (82.5% overall participation rate), though students were allowed to skip questions they did not want to answer. The analysed sample consists of 39,154 students from 1,807 classrooms and is representative of ninth graders in Germany (for details on the sampling process, see ref. 40).

The statistical power provided by this large dataset is particularly important for the study of social processes among refugee students, as they are still a comparatively small group (refugees constituted approximately 2.2% of the German population in 2020). For the analyses underlying both Figs. 2 and 3, we used subsamples of classrooms that were attended by at least one refugee student. In addition, Fig. 2 was restricted to classrooms where at least 15 students answered the questions about friendships and rejections. These samples comprised 5,328 students from 237 classrooms (Fig. 2) and 6,390 students from 304 classrooms (Fig. 3). Information on the samples and robustness analyses of various samples are provided in Supplementary Appendix B.

## Measurement

**Friendship and desk mate rejection.** Friendship and desk mate rejection were measured with a sociometric questionnaire. For friendship, students were asked, 'Who are you friends with?'; for rejection, they were asked, 'Who would you not want to sit next to?' (for an excerpt of the student questionnaire, see S6 in Supplementary Appendix C). While the latter question does not assess the full scope of social rejection, it captures a key aspect of rejection given the high importance of desk mate relations in adolescence[57].

**Immigrant and refugee status.** Students were asked about their country of birth as well as the country of birth of their parents and grandparents. School administrators provided information about the refugee status of students. From this, we derived a variable representing a student's immigration status. Refugee students were identified on the basis of the information provided by school administrators and were restricted to those who arrived in or after 2014 from Syria (60% of the refugee adolescents in our sample), Afghanistan (27%), Iraq (12%) or Lebanon (1%). These students typically receive one of three forms of legal protection granted to forced migrants in Germany: entitlement to asylum, refugee protection or subsidiary protection. Other students born outside of Germany were coded as first-generation immigrants. Those who were born in Germany but had at least one parent born outside of Germany were coded as second-generation immigrants. The rest of the students were categorized as native students. For the multivariate models, we created dyadic variables based on the immigrant status of the sender and the receiver of a friendship or rejection tie in each possible combination. We included all possible combinations of the immigrant status of the relationship sender and receiver in the model (16 combinations), except for the dyad native → native, which served as a reference category.

**Classroom ethnic diversity.** We calculated classroom diversity indices using the dissimilarity (or fractionalization) index[36,37,58,59]. This index expresses the number of pairs of students who have different countries of origin compared with the total number of student pairs in the classroom. The country of origin is the student's birthplace for first-generation immigrant students and the parents' birthplace for second-generation immigrant students. Second-generation immigrant students whose parents were born in different countries other than Germany were assigned to the category 'other origin'. For native students, Germany was considered the country of origin. In Fig. 2, low-diversity classrooms include diversity indices under 0.46, those of medium-diversity classrooms range from 0.46 to 0.76 and high-diversity classroom indices range above 0.76. Each of the three diversity settings constitutes one-third of the whole sample.

**Control variables.** We used information from the student questionnaire about the date the students first arrived in Germany (if they were born elsewhere) and about their country of origin. Information on students' age and gender was provided by school officials. Academic achievement was operationalized with students' grades in mathematics as reported by the schools. In Germany, grades range from 1 (excellent) to 6 (insufficient), yet we reversed this variable so that higher values indicate better achievement. The results from C-tests were used as indicators of general German language abilities. C-tests—a specific form of cloze tests—consist of a short text in which the second half of every second word is deleted. The students' task is to fill in the gaps[60]. These measures are of vital importance because refugees, on average, attain lower achievement levels than their classmates[54]. Controlling for academic achievement and German language skills allows us to rule out the possibility that the rejection of refugee students as desk mates reflects the rejection of low-achieving peers or is due to language barriers.

**Transforming individual variables into dyadic variables.** To examine how immigrant status is associated with social ties in multivariate social network models, we created dyadic variables from the immigrant status information. As control variables, we considered the characteristics of the tie sender and the tie receiver to determine whether students with certain attributes nominate others more often (sender effects) or are named by others more often (receiver effects). These variables account for the fact that many student characteristics are confounded with refugee status (for example, refugee students have lower levels of school achievement on average) and could affect their social acceptance. Additionally, we controlled whether pairs of students had the same country of origin, the same gender, a similar age, a similar length of stay in Germany, similar language skills and similar achievement. This accounts for the fact that students with the same immigrant status are often also similar in other ways (for example, language skills) and may thus cluster together in social networks due to homophily principles[61].

## Statistical analysis

The multivariate analyses (results shown in Fig. 3) aim to determine the statistical significance of associations between students' immigrant status and their social ties, while accounting for differences in the opportunity structure. Standard statistical methods, such as regression, cannot be used for this purpose because they assume that observations are independent of each other[62]. Due to endogenous processes such as reciprocity, clustering and self-reinforcing popularity in social networks[63–65], the independency assumption does not hold. To deal with endogeneity, various families of social network models have been developed[66].

We applied the Multiple Regression Quadratic Assignment Procedure (MRQAP), a linear regression framework for network data[46], which is one of the model families that accounts for the inter-dependencies within networks and, thus, network endogeneity[66]. MRQAPs are similar to linear probability models with non-parametric null distributions for standard errors[67]. The models control for the network structure (that is, the amount and distribution of mutual ties or clustering in the

network due to endogenous processes of reciprocity and transitivity) by comparing observed networks with simulated random networks with the same structure as a baseline. This is done by a permutational procedure, in which rows and columns of the matrix (that is, the actors/nodes) are simultaneously permuted in a way that leaves the network structure intact[62,66]. This way, MRQAPs control for the exact dependency structure of the network and account for the possibility that nominations between people with certain attributes (for example, native → native) may appear more likely purely because of the structure of the network. We apply the multi-group version of MRQAPs, which allows for the joint modelling of multiple networks (that is, all classrooms) together and has been applied in a number of social network studies of young peoples' social-tie preferences for example, refs. [68,69]. Multi-group MRQAPs restrict permutations to within-classroom dyads and ignore the substantively meaningless between-classroom dyads[69]. Details on the method and its mathematical foundations can be found in Supplementary Appendix D.

The dependent variables in our analyses were binary friendship and rejection variables between each pair of students. These variables had the value 1 if a friendship or rejection nomination existed and 0 if not. The independent variables were dummy or continuous variables representing characteristics of the tie senders, receivers and their match (for binary variables) or their similarity (for continuous variables). We included the main effect of ethnic diversity as well as interaction effects between ethnic diversity and the variables representing immigrant status in the model. Given the low number of refugee students in the majority of the classrooms, the estimation was performed jointly for all the classrooms to ensure sufficient statistical power[69].

## Strength of the analytical approach

Using social network data, this study measures how accepted refugee students actually are among their peers, instead of assessing how accepted they subjectively feel. Asking peers about social ties with classmates (and thus having two perspectives social relations among adolescents) provides a more valid measure of refugees' acceptance than asking about attitudes towards refugees, which may be susceptible to social desirability bias. In addition, while most network studies in the context of education focus on friendship only, we investigate a complementary indicator of social integration: rejection. This is important because being rejected as desk mates may impact students' success and wellbeing even more than simply not having friends[70]. In addition, rejection nominations seem to be more susceptible to students' preferences and prejudices than friendship nominations (see the 'Discussion' section) and, hence, be more sensitive indicators to detect the assumed mechanisms.

We combine an ego-network approach for descriptive analysis with a whole-network approach for multivariate statistical modelling. We first show the overall social integration of refugees under different diversity conditions by presenting refugee ego networks (Fig. 2). Then, we employ multivariate statistical modelling to explain the social mechanisms that produce these ego networks, while taking into account various characteristics of refugees that typically play a role in social-tie choices (for example, age, language skills, academic achievement and so on; Fig. 3). The whole-network approach allows us to control for peer characteristics, the role of network processes (for example, reciprocity, transitivity and so on) and baseline differences in the social integration of adolescents on the basis of diversity. In this way, we can consider that classrooms with higher levels of diversity may provide all students with higher levels of social integration, not only refugee students. Combining a descriptive ego-network approach with whole-network-based statistical modelling enables us to show the overall social integration of refugee students in different diversity settings and provide a thorough insight into the social mechanisms behind such diversity-based differences.

For the cross-sectional analysis of (whole) social networks, a commonly applied model family is the Exponential Random Graph Model (ERGM), for example, in ref. 37. For modelling multiple networks (in our case, classrooms) together, studies typically use ERGMs following a two-step approach in which individual classrooms are analysed first, and then parameters are meta-analysed. However, MRQAPs have the advantage of allowing a one-step approach, which is more appropriate in the case of our data. Given the large number of nomination types we model (all possible combinations between natives, refugees, first-generation and second-generation immigrants), the relatively small classrooms and the low number of refugees and first-generation immigrants in most classrooms, we do not have sufficient statistical power for classroom-level models, which would be the first step of a two-step ERGM approach. Another advantage of MRQAPs is that we can straightforwardly interpret the parameters as (additional) likelihoods for social ties to exist. In contrast, parameter interpretation is more difficult in the case of ERGMs[69].

## Robustness checks

**Robustness checks for Fig. 1.** We replicated the results of Fig. 1 in two different ways. First, we only included classrooms that were used for the analysis for Fig. 3 (Supplementary Fig. 1 in Supplementary Appendix B). Second, we estimated the same nomination tendencies after controlling for refugee adolescents' gender, age, academic achievement, language skills and length of stay in Germany (Supplementary Table 1 in Supplementary Appendix A). Both analyses yielded similar results to our main results (Supplementary Appendices A and B).

**Robustness checks for Fig. 2.** We excluded classrooms with fewer than 15 valid observations from Fig. 2 because the social networks of very few students could represent a biased number of friends among refugees. As a robustness check, we first recreated Fig. 2 using the same sample as in Fig. 3 (that is, the Fig. 2 sample without the size-related restrictions; Supplementary Fig. 2 in Supplementary Appendix B). Second, we replicated our findings using three alternative sets of diversity thresholds. For Supplementary Fig. 3 in Supplementary Appendix B, we used the lowest, medium and highest third of the diversity distribution of the complete instead of the analysed sample. For Supplementary Fig. 4 in Supplementary Appendix B, we relied on fixed thresholds of 0.33 (between low- and medium-diversity settings) and 0.66 (between medium- and high-diversity settings). For Supplementary Fig. 5 in Supplementary Appendix B, we calculated the lowest, medium and highest third of the diversity distribution of the analysed sample using immigrant proportion instead of the dissimilarity index as a measure of diversity. In each robustness check, the substantive findings are in line with the main results.

**Robustness checks for Fig. 3.** To account for the nested nature of the data, we first conducted a robustness check in which we included classroom fixed effects. These results are presented in Supplementary Appendix E, Supplementary Table 6. Second, we tested whether a different definition of classroom diversity provides similar results as the main model. This robustness model used the proportion of immigrants instead of our original diversity measure (Supplementary Appendix E, Supplementary Table 7). Third, to rule out that the results are due to geographical differences in diversity instead of classroom-level differences, we included diversity measures (and their interactions with each nomination type) at the classroom and federal-state levels. For diversity at the federal-state level, we relied on data from 2018's proportion of immigrants in each federal state[71]. At the classroom level, we included the proportion of immigrants as the diversity measure to maximize comparability. The results are presented in Supplementary Table 8 in Supplementary Appendix E. Fourth, to consider that students with higher socio-economic status may be more (or less) likely to name refugee students as friends or reject them as classmates, independent of their own

immigrant status, we controlled for individual socio-economic status of the tie sender and the interaction between sender's socio-economic status and the refugee status of the receiver (Supplementary Appendix E, Supplementary Table 9). Finally, we considered that students in different school tracks of the German education system might also be more or less likely to name refugee friends or reject them, independent of their own immigrant status. Therefore, additional models controlled for the attended school track and the interaction between the attended school track and the refugee status of the tie receiver (Supplementary Appendix E, Supplementary Table 10). This is important because, in Germany, the secondary education system is organized into different school tracks, which differ in their socio-economic composition and the students' achievement and language skills.

The robustness analyses overall confirm our substantive findings and conclusions (see more about the specific results in Supplementary Appendix E). In particular, the results imply that our main findings cannot be attributed to geographical instead of classroom-level variation in ethnic diversity, individual differences in socio-economic status or the attended school track. It should be noted, however, that the effect of diversity on native → refugee friendship nominations does not seem to be robust across different model specifications (that is, it remains positive but is not significant in each model). This implies that the positive diversity effect on refugee students' social integration, which emanates from changes in the behaviour of native peers, is mainly a result of reduced rejections, whereas the effect of diversity on natives' likelihood to befriend refugees is less clear.

### Reporting summary

Further information on research design is available in the Nature Portfolio Reporting Summary linked to this article.

## Data availability

The dataset analysed in the current study was made available for non-commercial research upon application at the Research Data Centre (FDZ) at the Institute for Educational Quality Improvement (IQB) (https://www.iqb.hu-berlin.de/fdz/studies/IQB-BT_2018). For this study, a preliminary internal version was analysed, which is available from the authors after signing a confidentiality agreement. The two datasets differ in terms of variable names and documentation.

## Code availability

Custom code supporting this study's findings is available on the Open Science Framework (https://osf.io/as38f/).

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

## Acknowledgements

The authors acknowledge financial support from the Volkswagen Foundation (project ISONET, grant number 93 489), the Economic and Social Research Council (project MiSoC, grant number ES/S012486/1), the Federal Ministry of Education and Research (project MuHiK, grant number 01JG2106), the Jacobs Foundation and the Alexander von Humboldt Foundation. The funders had no role in study design, data collection and analysis, decision to publish or preparation of the manuscript. The responsibility for the content of this publication lies with the authors.

## Author contributions

Z.B. and G.L. contributed equally to this work. Z.B., G.L., A.E. and M.J. designed the study. P.S., M.J. and G.L. were responsible for the data collection. Z.B. and G.L. analysed the data. Z.B., G.L. and A.E. wrote the paper with contributions and comments from all authors.

## Funding

## Competing interests

The authors declare that there are no conflicts of interest.

## Additional information

**Correspondence and requests for materials** should be addressed to Zsófia Boda or Georg Lorenz.

# Reporting Summary

## Statistics

For all statistical analyses, confirm that the following items are present in the figure legend, table legend, main text, or Methods section.

| n/a | Confirmed | |
|---|---|---|
| ☐ | ☒ | The exact sample size (*n*) for each experimental group/condition, given as a discrete number and unit of measurement |
| ☐ | ☒ | A statement on whether measurements were taken from distinct samples or whether the same sample was measured repeatedly |
| ☐ | ☒ | The statistical test(s) used AND whether they are one- or two-sided<br>*Only common tests should be described solely by name; describe more complex techniques in the Methods section.* |
| ☐ | ☒ | A description of all covariates tested |
| ☐ | ☒ | A description of any assumptions or corrections, such as tests of normality and adjustment for multiple comparisons |
| ☐ | ☒ | A full description of the statistical parameters including central tendency (e.g. means) or other basic estimates (e.g. regression coefficient) AND variation (e.g. standard deviation) or associated estimates of uncertainty (e.g. confidence intervals) |
| ☐ | ☒ | For null hypothesis testing, the test statistic (e.g. *F*, *t*, *r*) with confidence intervals, effect sizes, degrees of freedom and *P* value noted<br>*Give P values as exact values whenever suitable.* |
| ☒ | ☐ | For Bayesian analysis, information on the choice of priors and Markov chain Monte Carlo settings |
| ☐ | ☒ | For hierarchical and complex designs, identification of the appropriate level for tests and full reporting of outcomes |
| ☒ | ☐ | Estimates of effect sizes (e.g. Cohen's *d*, Pearson's *r*), indicating how they were calculated |

*Our web collection on statistics for biologists contains articles on many of the points above.*

## Software and code

Policy information about availability of computer code

| Data collection | No software was used. |
|---|---|
| Data analysis | The data were analysed using the open source software R (R Core Team, 2019). Custom code supporting this study's findings is available on the Open Science Framework (https://osf.io/as38f/). |

For manuscripts utilizing custom algorithms or software that are central to the research but not yet described in published literature, software must be made available to editors and reviewers. We strongly encourage code deposition in a community repository (e.g. GitHub). See the Nature Portfolio guidelines for submitting code & software for further information.

## Data

Policy information about availability of data

All manuscripts must include a data availability statement. This statement should provide the following information, where applicable:
- Accession codes, unique identifiers, or web links for publicly available datasets
- A description of any restrictions on data availability
- For clinical datasets or third party data, please ensure that the statement adheres to our policy

The dataset analyzed in the current study was made available for non-commercial research upon application at the Research Data Centre (FDZ) at the Institute for Educational Quality Improvement (IQB) (https://www.iqb.hu-berlin.de/fdz/studies/IQB-BT_2018). For this study, a preliminary internal version was analyzed, which is available from the authors after signing a confidentiality agreement. The two data sets differ in terms of variable names and documentation.

# Field-specific reporting

Please select the one below that is the best fit for your research. If you are not sure, read the appropriate sections before making your selection.

☐ Life sciences   ☒ Behavioural & social sciences   ☐ Ecological, evolutionary & environmental sciences

For a reference copy of the document with all sections, see nature.com/documents/nr-reporting-summary-flat.pdf

# Behavioural & social sciences study design

All studies must disclose on these points even when the disclosure is negative.

| | |
|---|---|
| Study description | This is a quantitative study using cross-sectional social network and survey data. |
| Research sample | The research sample stems from the Trends in Student Achievement Study 2018, conducted by the Institute for Educational Quality Improvement (IQB). The consists of 39,154 students from 1,807 classrooms and is representative of 9th graders at the country level (i.e., for Germany), the federal-state level, and the school-type level. Descriptive statistics of the analyzed variables by immigrant status, including age, sex, and immigrant background, are provided in Supplementary Table 3 in in SI Appendix B. The sample was chosen because it is the largest dataset on refugee students' social networks currently available. Additionally, the data set is representative of 9th graders in Germany. |
| Sampling strategy | The sample was selected by randomly drawing schools based on the distribution of secondary school types (e.g., academic track, intermediate track, and comprehensive track) in each federal state of Germany. Subsequently, classrooms were randomly drawn in each school (one in academic track secondary schools and two in all other schools). For details on the sampling process, see Stanat et al., 2019). <br><br> Reference: <br> Stanat, P., Schipolowski, S., Mahler, N., Weirich, S. & Henschel, S. IQB Trends in Student Achievement 2018. The Second National Assessment of Mathematics and Science Proficiencies at the End of Ninth Grade. Summary. (Waxmann, 2019a). |
| Data collection | By the decision of the Standing Conference of the Ministers of Education and Cultural Affairs of the States in the Federal Republic of Germany, participation in the proficiency tests for the IQB Trends in Student Achievement 2018 was compulsory both for students at public schools. The tests were conducted by the International Association for the Evaluation of Educational Achievement (IEA Hamburg). Additionally, paper-pencil questionnaires were handed out by IEA-Hamburg staff and filled out by students, teachers, parents, and school administrators. School administrators provided information about the refugee status of students. The researchers were blind to the study hypotheses during data collection. <br><br> Reference: <br> Stanat, P., Schipolowski, S., Mahler, N., Weirich, S. & Henschel, S. IQB Trends in Student Achievement 2018. The Second National Assessment of Mathematics and Science Proficiencies at the End of Ninth Grade. Summary. (Waxmann, 2019a). Available online at https://www.iqb.huberlin.de/bt/BT2018/Bericht |
| Timing | The data were collected between April 23 and June 22, 2018 (for details, see Stanat et al. 2019). <br><br> Reference: <br> Stanat, P., Schipolowski, S., Mahler, N., Weirich, S. & Henschel, S. IQB Trends in Student Achievement 2018. The Second National Assessment of Mathematics and Science Proficiencies at the End of Ninth Grade. Summary. (Waxmann, 2019a). Available online at https://www.iqb.huberlin.de/bt/BT2018/Bericht |
| Data exclusions | The analyzed sample consists of 39,154 students from 1,807 classrooms. For the analyses underlying Fig 3, we used subsamples of classrooms that were attended by at least one refugee student (Fig 3) and where at least 15 students answered the questions about friendships and rejections (Fig 2). These samples comprised 6,390 students from 304 classrooms and 5,328 students from 237 classrooms. Information on these samples and robustness analyses of various samples are provided in SI Appendix B. |
| Non-participation | Participation in the achievement tests was mandatory in all public schools, resulting in a participation rate of 92.4%. Completing the student questionnaire was voluntary in some federal states and mandatory in others, requiring parental consent in the latter states (82.5% overall participation rate). |
| Randomization | The participants were not allocated into experimental groups. |

# Reporting for specific materials, systems and methods

We require information from authors about some types of materials, experimental systems and methods used in many studies. Here, indicate whether each material, system or method listed is relevant to your study. If you are not sure if a list item applies to your research, read the appropriate section before selecting a response.

## Materials & experimental systems

| n/a | Involved in the study |
|-----|----------------------|
| ☒ ☐ | Antibodies |
| ☒ ☐ | Eukaryotic cell lines |
| ☒ ☐ | Palaeontology and archaeology |
| ☒ ☐ | Animals and other organisms |
| ☐ ☒ | Human research participants |
| ☒ ☐ | Clinical data |
| ☒ ☐ | Dual use research of concern |

## Methods

| n/a | Involved in the study |
|-----|----------------------|
| ☒ ☐ | ChIP-seq |
| ☒ ☐ | Flow cytometry |
| ☒ ☐ | MRI-based neuroimaging |

# Human research participants

Policy information about studies involving human research participants

| | |
|---|---|
| Population characteristics | Descriptive statistics of the human research participants are provided in SI Appendix B. In the research sample, 49 % of the particpants were female and the average age was 15.6. The classroom social networks contained 24.9 students, on average. |
| Recruitment | See above. For details on the sampling process, see Stanat, P., Schipolowski, S., Mahler, N., Weirich, S. & Henschel, S. IQB Trends in Student Achievement 2018. The Second National Assessment of Mathematics and Science Proficiencies at the End of Ninth Grade. Summary. (Waxmann, 2019a). Available online at https://www.iqb.huberlin.de/bt/BT2018/Bericht |
| Ethics oversight | The data collection was part of the educational monitoring strategy ratified by the Standing Conference of the Ministers of Education and Cultural Affairs of the Länder in the Federal Republic of Germany. The German school laws and regulations state that participation in large-scale school assessment studies aiming to assure educational quality (including the IQB Trends in Student Achievement studies, but also the PISA, TIMSS, and PIRLS studies) can be obligatory for schools, school principals, teachers, and students. In accordance with these laws and regulations, the study participants are informed about the general content of the achievement tests and surveys in advance of this monitoring. The Ministries of Education of each federal state approve the data collection, including all of the instruments. This approval procedure considers ethical aspects as well as data protection requirements according to German law. For the 2018 Trends in Student Achievement study (which provides the data used in the present study), the Ministries of Education agreed to waive consent requirements and endorse compulsory participation for the above reasons. However, as school laws differ between the 16 federal states, the exact procedure varied: While participation in the achievement test was mandatory in all states, participation in the questionnaires was mandatory in some states (although students were free to skip questions they did not want to answer) and voluntary in others. |

Note that full information on the approval of the study protocol must also be provided in the manuscript.

