## [Peer Review File · Nature Human Behaviour]

Peer Review Information

Journal: Nature Human Behaviour

Manuscript Title: Ethnic diversity fosters the social integration of refugee students

Corresponding author name(s): Zsófia Boda, Georg Lorenz

Reviewer Comments & Decisions:

Decision Letter, initial version:

2nd March 2022

Dear Dr. Lorenz,

Thank you once again for your manuscript, entitled "Ethnic diversity fosters the social integration of refugee students", and for your patience during the peer review process. I am very sorry for the long delay in reaching a decision.

Your Article has now been evaluated by 2 referees, whose comments you will find included at the end of this letter. We also recruited a third reviewer, but unfortunately, they have not yet returned comments. If we do receive feedback from this reviewer in the future, I will pass it along to you. You will see from the comments of the two reviewers that, although they find your work of potential interest, they have raised quite substantial concerns. In light of these comments, we cannot accept the manuscript for publication, but would be interested in considering a revised version if you are willing and able to fully address reviewer and editorial concerns.

We hope you will find the referees' comments useful as you decide how to proceed. If you wish to submit a substantially revised manuscript, please bear in mind that we will be reluctant to approach the referees again in the absence of major revisions. We are committed to providing a fair and constructive peer-review process. Do not hesitate to contact us if there are specific requests from the reviewers that you believe are technically impossible or unlikely to yield a meaningful outcome.

To guide the scope of the revisions, the editors discuss the referee reports in detail within the team, including with the chief editor, with a view to (1) identifying key priorities that should be addressed in revision and (2) overruling referee requests that are deemed beyond the scope of the current study. We hope that you will find the prioritised set of referee points to be useful when revising your study. Please do not hesitate to get in touch if you would like to discuss these issues further.

In particular, your revision must address the following (as well as all other reviewer comments):

- 1.) Ensure that your choice of methodology is fully explained and justified. We encourage you to consider estimating the ERGMs that Reviewer 2 recommends as an alternative approach or robustness check.
- 2.) Address concerns raised by both reviewers regarding the interpretation of results and the clarity of core concepts.
- 3.) Address Reviewer 2's concerns regarding the measurement of diversity and the exclusion of main effects in models containing interaction terms.

If you wish to submit a suitably revised manuscript we would hope to receive it within 4 months. We understand that the COVID-19 pandemic is causing significant disruptions which may prevent you from carrying out the additional work required for resubmission of your manuscript within this timeframe. If you are unable to submit your revised manuscript within 6 months, please let us know. We will be happy to extend the submission date to enable you to complete your work on the revision.

- Include a "Response to the editors and reviewers" document detailing, point-by-point, how you addressed each editor and referee comment. If no action was taken to address a point, you must provide a compelling argument. This response will be used by the editors to evaluate your revision and sent back to the reviewers along with the revised manuscript.
- Highlight all changes made to your manuscript or provide us with a version that tracks changes.

[REDACTED]

Thank you for the opportunity to review your work. Please do not hesitate to contact me if you have any questions or would like to discuss the required revisions further.

Sincerely,
Aisha

Aisha Bradshaw, PhD

Senior Editor
Nature Human Behaviour

Reviewer expertise:

Reviewer #1: did not return comments

Reviewer #2: interethnic relations, student social networks, network analysis methods

Reviewer #3: interethnic relations, social networks

REVIEWER COMMENTS:

Reviewer #2:
Remarks to the Author:

Key results

This study examines peer relations of adolescent refugees in German schools, with emphasis on how these are affected by ethnic diversity in school. Descriptively, the study shows that refugee students report fewer friends and are more often rejected as desk mates than both native majority and non-refugee ethnic minority peers. Using regression models, the study further shows that non-refugee immigrant-origin students are more accepting of refugee peers as desk mates than native origin youth are. Native-origin youth further are more likely to have refugees and friends and less likely to reject them as desk mates in schools with higher ethnic diversity. The study concludes that, as the title states, "ethnic diversity fosters the social integration of refugee students."

Validity

For both substantive and methodological reasons, I am not fully convinced of the validity of the data interpretation and conclusions. In this section I outline my substantive concerns; I discuss my methodological concerns in the "analytical approach" section below.

Setting aside methodological concerns for a moment, I am puzzled by the clear-cut policy recommendation of allocating refugees to high diversity settings to "promote their social adjustment and mitigate the negative consequences of prejudice and intergroup bias." As the manuscript states, if refugees attend high diversity schools, they have "more opportunities to meet other ethnic minority peers". Yet this is just simple math; we do not need an empirical study to make this statement. So, the question is whether preferences vary by diversity, i.e., whether non-refugee students in higher diversity settings are more positive towards refugees.

If we take the values in Figure 3 at face values, to "foster social integration", refugees would have to attend classrooms with diversity levels above .5 or even .75. But this can hardly be a serious policy

recommendation. In societies in which ethnic minority members make up less than half of the population, it is hardly possible to distribute refugees in settings with very high diversity levels. And even if one would be able to do it, wouldn't this lead to polarization, because it implies that high shares of native-origin students are educated in low diversity settings without the possibility of engaging with refugees and, thus, improving their attitudes towards them? Also, the study ignores that refugees in high diversity have, necessarily, less contact with the native majority group. Isn't it likely that refugees in low diversity settings who have more contact with the native majority group develop more positive attitudes towards the majority group than refugees in high diversity settings? Isn't this also part of "social integration"? (On p. 3 it is stated that "social integration is not a one-sided process but depends on the simultaneous attitudes and behaviors of multiple actors".)

Significance

The conclusions are potentially relevant for the field. That said, like many other studies on refugees, this study lacks new theoretical arguments, as it seems to assume that studying refugees is important by itself. Therefore, I would describe it as a test of established mechanisms and patterns for a subgroup of immigrants.

Data and methodology

The abstract and introduction emphasize the large dataset of "39,154 students in 1,807 classrooms". But the multivariate analysis on which the conclusions are based relies on only about 6,300 students in 300 classrooms. I understand why the number is smaller, and I do not think that this diminishes the study. But it seems a bit disingenuous to advertise the study with a much larger sample size. (As for the methodology, I do not understand how my assessment of the "methodology" would differ from my assessment of the "analytical approach"; so I do comment on methods in the next section.)

Analytical approach

The study uses the multiple regression quadratic assignment procedure (MRQAP). Noting that I am not an expert on these methods, I have several questions.

First, most studies analyzing cross-sectional network data use exponential random graph models (ERGMs). This includes studies that examine the relationship between individual preferences and diversity, such as the one by Smith et al. (2016) cited in the manuscript. As the manuscript says on p. 9, "endogenous processes such as reciprocity, clustering, and self-reinforcing popularity" matter for social networks. I understand that MRQAP should control for endogenous network effects, but can we be sure that they do so as well as ERGMs would? Since identifying preferences is crucial to this study, I also was wondering why it uses another method than most other studies in this area. I think this warrants some explanation. (And perhaps consider estimating ERGMs for a robustness check.)

Second, leaving aside the question of whether MRQAP are more suited than ERGMs, I was wondering about the model specification. The model underlying Figure 3 relies on interaction effects (immigrant status x diversity). On p. 10 it is said that "the main effect of ethnic diversity" is included, but Table S6 says that "the diversity main effect is excluded to avoid multicollinearity." Moreover, since diversity is interacted with "native -> refugee", "native -> 1st-gen immigrant" and so forth, wouldn't one also have to include these main effects, i.e., "native->refugee", "native-> 1st-gen immigrant" etc. At least

this is what one would have to do in a conventional regression model. Here, too, some explanation would be helpful.

Third, for the study, it is of utmost importance to identify preferences. But isn't the fact that refugees are, by definition, different from 1st and 2nd generation immigrants an obvious alternative explanation to the one offered in the manuscript? After all, the students in the analysis are in grade 9. Therefore, they have already attended their current school for two or even up to four years. Obviously, in these years' students formed friendships. But what if refugees just, on average, entered class more recently? Is this information available? I know that length of residence is controlled for, but this does not necessarily equal the number of years refugee students were part of the classroom. Fourth, why does Figure 3 do not include confidence intervals? Wouldn't this be helpful to assess differences between groups and by diversity?

Suggested improvements

Two suggestions follow from my previous comments on the analytical approach. First, either explain why MRQAP are better suited than ERGMs or consider estimating ERGMs. Second, either explain why the main effects of interaction terms are not included in the MRQAPs or include these effects. (In any case, be consistent, i.e., do not say on one page that an effect is included if a note in the appendix says that it is excluded.)

I do not see a strong reason to capture diversity by categories such as "low", "medium", and "high"; the underlying measure is metric. Moreover, if categories must be used, I suggest using different thresholds for defining "low", "medium", and "high" diversity. I am not sure how classrooms with about .5 diversity can reasonably be described as "low" diversity settings.

Clarity and context

The text is mostly clear and accessible. However, the key term "social integration" is nowhere defined. Worse, the term seems to be used in two different meanings. On the one hand, in some parts of the manuscript, "social integration" seems to refer to social contacts between refugees and the native majority. For example, in the first paragraph of the introduction, the authors stress the importance of "peer relationships across ethnic boundaries because social integration is a key determinant of success in the school system and beyond." This is how the term is typically used in research on immigrants' integration (e.g., by Gordon or Alba), and, thus, how it is used in many of the references cited in the introduction (references 9-12). On the other hand, it later becomes clear that the authors seem to use the term "social integration" in reference to *any* kind of social contacts (e.g., in the third paragraph of the introduction or when referring to the number of friends of any group as an indicator of social integration). Given that the term is key for the manuscript, I recommend defining it early on and then using it in an unequivocally way.

Irrespective of the ambiguity of the term "social integration", the manuscript distinguishes between "friendship and rejection networks". "Rejection" is measured by the question "Who would you not want to sit next to?" in class. I understand that this probably measures a negative social relationship, but I am still puzzled whether it is accurate to say that a student "rejects" another one if she does not want to sit next to her in class. After all, there might be strategic reasons to avoid a specific peer as desk mate, for example, because they are too loud or maybe even because one is friends with her and she

might thus be a distraction from studying. I am not saying that these examples are necessarily more accurate than the interpretation of the authors, but I do think that it is a bit of a stretch to conclude from this measure that "refugees are rejected". One solution might be to just always state "rejected as desk mates", another one would be to provide additional evidence that the item actually measures a genuine negative feeling towards another person.

The results are provided with enough context; the figures are very helpful. Previous work is considered sufficiently.

References

The manuscript appropriately references previous literature.

Your expertise

I am not familiar with MRQAP, which is the method the authors use for most of their analysis. I must admit that I do not understand why they chose this method, as they seem to fall behind the current state-of-the-art of analyzing cross-sectional data with exponential random graph models (ERGMs). I know that the authors are familiar with ERGMs, and ERGMs also were used in the studies that are arguably most similar to this study (e.g., the one by Smith et al. 2016, cited in the manuscript). While I do think that ERGMs are the more appropriate choice, perhaps there are good reasons for favoring MRQAP over ERGMs. But these are not explained in the manuscript or appendix.

Reviewer #3:

Remarks to the Author:

Review of the article "Ethnic diversity fosters the social integration of refugee students" for Nature Human Behaviour.

Using a unique large nationally representative data set of social networks of refugee, non-refugee minority, and majority students, this study is the first to show the social integration of refugee students. This provides novel insights into how refugee students can be best accommodated. The study also shows the importance of the ethnic diversity of the local context (classroom) for students' willingness to socialize with newcomers.

In my view, this study is of immediate interest to policymakers and scholars studying social integration and social cohesion in many disciplines. The available high-quality data and the excellent application of sophisticated methods for social network analysis give credibility to the results. The analyses are adequately described in the supplemental material and a series of robustness checks increase trust in the conclusions drawn. The article and abstract are clearly written and should be accessible to a wide audience.

I have only a few notes on what the authors could do to strengthen the paper further:

1. I'm uncertain about the meaning of the finding that "first-generation immigrant students are more likely to both befriend and reject their refugee peers in more diverse classrooms" (p.7). From Figure 3, we learn that the probability of rejection in high diversity classrooms is similar to that of other groups. However, first-generation immigrant students are less likely to reject refugees in the low-diversity classrooms. Importantly, there are relatively few first-generation students in the sample (472) and the logic of the analyses suggests that very few of them will be in low diversity classrooms. As a consequence, I wonder if the odd result of less rejection in low diversity classrooms is the consequence of a few extreme cases (outliers). An outlier analysis of the first-generation immigrant students would be reassuring that this finding is robust.

2. What about the academic school level (tracking)? Schools of a higher academic track tend to be less ethnically diverse than those of a lower track. Is what we see, in part, related to academic tracking?

Minor notes:

3. The results in Figure 2 are very interesting but they might also lead to a biased interpretation of readers. One could read them to say that refugee students are mainly rejected by native students and – as soon as there are sufficient minority students – mainly befriended by minority students. However, this does not take the baseline probability into account. In low diversity classes, there are mainly native students who could send rejection ties. And in high diversity classes, there are relatively fewer native students who could send friendship ties. I would suggest adding descriptive analyses that take the baseline probability into account (e.g. proportions) – next to the analysis that are now shown.

4. I would remove one "available" from this sentence (p. 4)
 "Our analyses are based on the largest available dataset on refugee students' social networks currently available."

Signed: Tobias Stark

Author Rebuttal to Initial comments

Reviewer #2:

Comment #2.1 (Validity)

Setting aside methodological concerns for a moment, I am puzzled by the clear-cut policy recommendation of allocating refugees to high diversity settings to "promote their social adjustment and mitigate the negative consequences of prejudice and intergroup bias." [...] So, the question is whether preferences vary by diversity, i.e., whether non-refugee students in higher diversity settings are more positive towards refugees.

If we take the values in Fig 3 at face values, to "foster social integration", refugees would have to attend classrooms with diversity levels above .5 or even .75. But this can hardly be a serious policy recommendation. In societies in which ethnic minority members make up less than half of the population, it is hardly possible to distribute refugees in settings with very high diversity levels. And even if one would be able to do it, wouldn't this lead to polarization, because it implies that high shares of native-origin

students are educated in low diversity settings without the possibility of engaging with refugees and, thus, improving their attitudes towards them? Also, the study ignores that refugees in high diversity have, necessarily, less contact with the native majority group. Isn't it likely that refugees in low diversity settings who have more contact with the native majority group develop more positive attitudes towards the majority group than refugees in high diversity settings? Isn't this also part of "social integration"? (On p. 3 it is stated that "social integration is not a one-sided process but depends on the simultaneous attitudes and behaviours of multiple actors".)

Thank you for these thoughtful comments that clearly helped us to reframe our study results' implications. First of all, we would like to note that our substantive conclusion (i.e., ethnic diversity fosters the social integration of refugees) follows from the multivariate models reported in Fig 3. These results allow us to identify differences in the tendencies of majority, non-refugee minority, and refugee adolescents to be friends with and/or reject refugee classmates based on the level of ethnic diversity. We now briefly address this issue in the first paragraph of the discussion section (page 7):

"However, in more ethnically diverse classrooms, refugee adolescents are socially better integrated: they tend to have more friends and are rejected significantly less frequently as desk mates than in less diverse classrooms. This is revealed in descriptive analyses and multivariate social network models. The latter control for various factors relevant to social network dynamics, among them academic achievement and German language skills, which may make refugees less desirable as desk mates for their peers even in the absence of actual dislike. Notably, refugee students' improved social integration in more diverse classroom is not solely due to the preferences and higher shares of ethnic minority students, but variation in the preferences of majority-group adolescents for befriending and rejecting refugee peers across social contexts. In more diverse contexts, majority-group adolescents reject refugees as desk mates less often and tend to nominate them as friends more often than in less diverse contexts."

Then, we agree that allocating refugees to more diverse social contexts is not a plausible policy recommendation, especially for countries and regions with significant variation in diversity levels. This also applies to Germany, where some regions are marked by low ethnic diversity. Refusing to allocate refugees to such regions might indeed reinforce or even create a polarization of positive and negative attitudes towards refugees and immigrants in the country. In contrast, our results require a more nuanced discussion of possible implications that we are providing in the new discussion section on pages 8 and 9. The respective paragraphs sound as follows:

"From a political perspective, the finding that ethnically diverse school settings provide better conditions for the social integration of refugees challenges critical views of multiculturalism. Our results imply that placing young refugees in school environments that are already ethnically diverse can, to some degree, promote their social adjustment due to social acceptance by other ethnic minority peers, but also due to a reduced rejection by majority-group members, who are more accepting of refugee peers in more diverse classrooms. Given that positive contact with majority-group peers is critical for immigrant students' academic success and school adaptation, assigning refugees to ethnically diverse schools and classrooms might also foster their economic integration and future life opportunities (Lorenz, Boda, Salikutluk, et al. 2021).

Deducing policy recommendations from the study's findings is particularly difficult for countries and regions with significant variation in diversity levels. Taken at face value, the results suggest that it would be best for refugee students to attend highly diverse schools. However, policy advice needs to take various outcomes into account, including student academic achievement. Refugee students' language development, for instance, benefits from a high proportion of language majority students in the attended school classes, and it is a crucial determinant of their academic achievement (Schipolowski et al. 2021). Moreover, steering refugee students into more diverse schools would increase segregation, with some schools being attended by high proportions and others by only very few or no minority students. In such a scenario, intergroup contact would continue to be low in contexts marked by low diversity, and the processes allowing for more positive intergroup attitudes and higher social integration reported in this study would hardly occur there. To eventually blur ethnic boundaries, it would thus be important to ensure that diversity spreads out. Our findings clearly show that, in this process, special attention must be paid to refugee students in low-diversity settings. They seem particularly vulnerable to not being socially accepted in contexts with low levels of diversity, and institutional support for their social integration is vital. Therefore, school principals and teachers need to be aware of these challenges and support integration by, among other things, encouraging cooperation, setting up common goals, and showing explicit support for mixing ethnic groups (Pettigrew and Tropp 2006)."

Comment #2.2 (Significance)

The conclusions are potentially relevant for the field. That said, like many other studies on refugees, this study lacks new theoretical arguments, as it seems to assume that studying refugees is important by itself. Therefore, I would describe it as a test of established mechanisms and patterns for a subgroup of immigrants.

We agree that our study does not develop any new theoretical arguments. Instead, we are exploring the social integration of newly arrived refugees based on general theoretical notions from sociology and social psychology and test to what extent the diversity in the receiving contexts makes a difference in the social acceptance and social rejection of young refugees. The arrival of such a high number of new immigrants into the school system who are then randomly distributed across contexts provides an excellent opportunity to test such predictions. Further, even though we agree that a study is not important per se just because refugees are studied, the arrival of refugees in Western Europe in the last decade was still one of the most significant and controversial political topics. The current influx of refugees from the Ukraine stresses that the topic will most likely remain a key societal issue. Thus, we would argue that this is a group of particular interest, which also systematically differs from other migrants (e.g., in terms of migration motives, language skills upon arrival, etc.), and empirical results on which policy insights can be based are needed.

Comment #2.3 (Data and methodology)

The abstract and introduction emphasize the large dataset of "39,154 students in 1,807 classrooms". But the multivariate analysis on which the conclusions are based relies on only about 6,300 students in 300 classrooms. I understand why the number is smaller, and I do not think that this diminishes the study. But it seems a bit disingenuous to advertise the study with a much larger sample size.

We agree and changed the abstract accordingly. Although part of our analysis (i.e., Fig 1) is based on the full sample, we now highlight the number of refugee adolescents and their classmates in the abstract. The respective sentence sounds as follows:

"Using a large, nationally representative social-network dataset from Germany, we examine the relationships of refugee adolescents with their peers (304 classrooms, 6,390 adolescents, 487 refugees)."

In the introduction section, we kept the information on the size of the full sample because this sample served as the empirical bases for Fig 1 (comparison of refugees and their peers). However, we now describe the number of refugee adolescents and the number of classrooms they attended during the survey and highlight that our analysis of social integration is based on these individuals (page 4):

"The full data include complete friendship and desk-mate-rejection networks of 39,154 secondary school students in 1,807 school classes in Germany, with 342,114 friendships and 161,430 rejection relations measured. In the data, we identified 487 refugee adolescents in 304 classrooms, including 6,390 students in total."

Comment #2.4 (Analytical approach)

a) The study uses the multiple regression quadratic assignment procedure (MRQAP). Noting that I am not an expert on these methods, I have several questions.

First, most studies analyzing cross-sectional network data use exponential random graph models (ERGMs). This includes studies that examine the relationship between individual preferences and diversity, such as the one by Smith et al. (2016) cited in the manuscript. As the manuscript says on p. 9, "endogenous processes such as reciprocity, clustering, and self-reinforcing popularity" matter for social networks. I understand that MRQAP should control for endogenous network effects, but can we be sure that they do so as well as ERGMs would? Since identifying preferences is crucial to this study, I also was wondering why it uses another method than most other studies in this area. I think this warrants some explanation. (And perhaps consider estimating ERGMs for a robustness check.)

Indeed, several studies conducted cross-sectional analyses of the relationship between diversity and interethnic social ties using two-step ERGMs. Yet, we opted for a different approach. We will subsequently explain 1) why we chose a one-step approach instead of a two-step approach and 2) why the QAP framework (which made a one-step approach possible) is the more suitable method for our purposes.

Previous articles using ERGMs in the school context followed a two-step approach, in which researchers first estimated a model for each individual classroom and then further analyzed the overall estimates in a second step. We note, though, that most of these studies differentiated between only relatively few types

of social ties. Smith et al. (2016), for instance, distinguished between three types: two kinds of same-ethnic ties (both native; both immigrant from the same country) and inter-ethnic ties. This allowed them to estimate the same model for each classroom, as they had enough observations for each of nomination type (though they still had to exclude a number of classrooms from the analysis). In contrast, we differentiate between 16 different nomination types (all possible combinations between natives, refugees, first-generation immigrants, and second-generation immigrants). This means that estimating the same classroom-level model (be it ERGM or QAP) would not be possible for any classrooms that has at most one member of any of the four groups we examine (for instance, if there is one refugee in a classroom, no refugee→refugee ties could exist as one cannot nominate oneself). This would lead to the exclusion of the majority of our sample (234 out of 304 classrooms), partly because many classrooms have only one refugee student in the first place. Even if we estimated a somewhat different model for each classroom (excluding the variables for which we have no observations in the given classroom), it would be highly unlikely that we yield significant results for the nomination of refugees by other students, which is our main research aim. This is due to the low number of refugees in most classrooms (for instance, with 1 refugee and 3 first-generation immigrants, we would still only have 3 potential observations to estimate first-generation-immigrant→refugee friendship tendencies, but even with 1 refugee and 10 natives, we would have 10 potential observations only). We note that this issue would disproportionately affect classrooms with different diversity levels: in lower-diversity settings, it would be difficult to find significant results for nominations by immigrant-background students (due to their low number in these settings), whereas in higher-diversity settings, we would expect fewer significant results about nominations by natives than in less diverse settings. This could cause strong bias in our analyses.

In contrast to ERGMs, which require a two-step approach when analyzing several social networks, a multigroup option has been developed for QAPs. This option is able to handle networks from multiple classrooms together. Multigroup QAPs (MRQAPs) have been used in a number of social network studies recently, and even though these studies are not related to ethnicity, they investigate young peoples' social-tie preferences (e.g., Burnett-Heyes et al., 2015, Elmer and Stadtfeld 2020). Both MRQAPs and ERGMs account for network dependencies, though there is an important difference in how they do so. As described in Snijders (2011), ERGMs model the network structure by including specific parameters to capture aspects of tie configurations. At the same time, QAPs control for the network structure by comparing the observed networks to simulated random networks with the same structure as a baseline. This is done by a permutational procedure, in which rows and columns of the matrix (that is, the nodes) are simultaneously permuted in a way that leaves the network structure intact (Snijders 2011). This way, QAPs control for the exact dependency structure of the network. Given that we do not have research questions specifically concerning structural variables, such as reciprocity or transitivity, controlling for the network structure instead of explicitly modeling it is satisfactory for our analysis. An added advantage of MRQAPs is that we are able to interpret parameter sizes, which is difficult in the case of ERGMs.

We now briefly explain our choice of methodology in the new manuscript in the sections "Statistical analysis" and "Strength of the analytical approach" (see pages 11 and 12). The text reads as follows:

"We applied the Multiple Regression Quadratic Assignment Procedure (MRQAP), a linear regression framework for network data (Dekker et al. 2007), which is one of the model families that accounts for the interdependencies within networks and, thus, network endogeneity (Snijders, 2011). MRQAPs are similar to linear probability models with nonparametric null distributions for standard errors (see Wooldridge 1999, section 7.5). The models control for the network structure (i.e., the amount and distribution of mutual ties or clustering in the network due to endogenous processes of reciprocity and transitivity) by comparing observed networks to simulated random networks with the

same structure as a baseline. This is done by a permutational procedure, in which rows and columns of the matrix (i.e., the actors/nodes) are simultaneously permuted in a way that leaves the network structure intact (Snijders 2011). This way, MRQAPs control for the exact dependency structure of the network and account for the possibility that nominations between people with certain attributes (e.g., native → native) may appear more likely purely because of the structure of the network. The multigroup version of MRQAP allows for the joint modelling of multiple networks (i.e., all classrooms) together and has been applied in a number of social network studies of young peoples' social-tie preferences (e.g., Burnett Heyes et al. 2015; Elmer and Stadtfeld 2020). Details on the method and its mathematical foundations can be found in SI Appendix D.

[...]

For cross-sectional social network analysis, a commonly applied model family is the Exponential Random Graph Model (ERGM) (e.g. Smith et al. 2016). For modeling multiple networks (in our case, classrooms) together, studies typically use ERGMs following a two-step approach in which individual classrooms are analyzed first, and then parameters are meta-analyzed. However, MRQAPs have the advantage of allowing a one-step approach, which is more appropriate in the case of our data. Given the large number of nomination types we model (all possible combinations between natives, refugees, first-generation, and second-generation immigrants), the relatively small classrooms, and the low number of refugees and first-generation immigrants in most classrooms, we do not have sufficient statistical power for classroom-level models, which would be the first step of a two-step ERGM approach. Another advantage of MRQAPs is that we can straightforwardly interpret the parameters as (additional) likelihoods for social ties to exist. In contrast, parameter interpretation is more difficult in the case of ERGMs (Elmer and Stadtfeld 2020)."

Comment #2.5

b) Second, leaving aside the question of whether MRQAP are more suited than ERGMs, I was wondering about the model specification. The model underlying Figure 3 relies on interaction effects (immigrant status x diversity). On p. 10 it is said that "the main effect of ethnic diversity" is included, but Table S6 says that "the diversity main effect is excluded to avoid multicollinearity." Moreover, since diversity is interacted with "native -> refugee", "native -> 1st-gen immigrant" and so forth, wouldn't one also have to include these main effects, i.e., "native->refugee", "native-> 1st-gen immigrant" etc. At least this is what one would have to do in a conventional regression model. Here, too, some explanation would be helpful.

We agree with your considerations. However, there seems to be a misunderstanding here. The complete results corresponding to Fig 3 are presented in Table S2. Table S6, in turn, displays the results of the classroom-fixed-effect MRQAP models that we estimated as part of our robustness checks. In fact, the analyses underlying Fig 3 (displayed in Table S2) included the main effect of ethnic diversity. We changed the layout of Table S2 to make this more obvious for the reader.

The diversity main effect is excluded only from the statistical models shown in Table S6. These are classroom-fixed-effect MRQAP models serving as a robustness check. To avoid multicollinearity with the classroom dummy variables, we had to exclude the diversity main effect from these models.

Comment #2.6

c) Third, for the study, it is of utmost importance to identify preferences. But isn't the fact that refugees are, by definition, different from 1st and 2nd generation immigrants an obvious alternative explanation to the one offered in the manuscript? After all, the students in the analysis are in grade 9. Therefore, they have already attended their current school for two or even up to four years. Obviously, in these years' students formed friendships. But what if refugees just, on average, entered class more recently? Is this information available? I know that length of residence is controlled for, but this does not necessarily equal the number of years refugee students were part of the classroom.

We agree that longer exposure to classmates might be a source of more friendships (and possibly fewer rejections), although the assumption for rejections could also go in the opposite direction (i.e., more exposure induces a higher likelihood of rejections due to familiarity, see Zajonc [1968]). Consequently, refugees who had attended their classes for a shorter time period during the time of the data collection might have fewer friends than other refugee students and other ethnic minority adolescents, particularly those born in Germany. We only categorize students as refugees if they arrived in Germany in or after 2014, therefore, they have indeed likely spent less time in their school than other first-generation immigrants.

Unfortunately, the data provide no information on when the refugee (or any other) students first entered their classes. However, we highlight that a possible exposure mechanism would be unlikely to affect our substantive conclusion that ethnic diversity has a positive effect on the social integration of refugee adolescents. Due to the random allocation of refugees according to governmental quota (Königssteiner Schlüssel), the time point of refugees' arrival in the school classes is unconfounded with the ethnic diversity in the school classes.

We acknowledge these points as part of the interpretation of our results presented in Fig 1. In the revised manuscript, on page 8, we write the following:

"Another factor that might contribute to lower levels of social integration of refugees is that they have joined the classroom later than other students. Our multivariate models control for how long each student has been in Germany but not how long they have been in the classroom. Our data provide no information on when the refugee students had first entered their classes. Therefore, we cannot rule out that less time spent in the classroom (beyond less time spent in the country) partially explained the lower levels of social integration of refugees. However, this mechanism is unlikely to influence the role of diversity in this process."

Comment #2.7

d) Fourth, why does Figure 3 do not include confidence intervals? Wouldn't this be helpful to assess differences between groups and by diversity?

Thank you for this suggestion. Our model uses a permutations-based test to assess statistical significance. This procedure tests if each parameter is larger in absolute value than what we would expect if we estimated this parameter from random networks with the same structure. Therefore, traditional confidence intervals are not applicable in the QAP framework. Instead, we looked at the distribution of each parameter value calculated from a 1000 simulated ("random") networks and plotted where our estimated parameter value from the observed network falls within this distribution. If the parameter value from the observed network is higher/lower than 95% of the distribution of the same parameter from the simulated networks, the parameter is deemed statistically significant. In line with this, we have now included figures for each parameter plotted in Fig 3, included in Figure S7 in SI Appendix F. The dots in these figures represent the observed parameter values. The boxes represent the same parameter values in the simulated networks and reflect parameter sizes we would expect if the immigrant status of the dyads did not matter. The boxes and error bars together account for 95% of the distribution (from 1,000 simulations). The further away the dot is from the box, the least likely we would observe the given parameter by chance only.

Comment #2.8 (Suggested improvements)

1. Two suggestions follow from my previous comments on the analytical approach. First, either explain why MRQAP are better suited than ERGMs or consider estimating ERGMs. Second, either explain why the main effects of interaction terms are not included in the MRQAPs or include these effects. (In any case, be consistent, i.e., do not say on one page that an effect is included if a note in the appendix says that it is excluded.)

The justification of ERGMs can be found in our response to your Comment #2.4. Our explanation about diversity main effects is included in our response to your Comment #2.5. We describe the MRQAP procedure in more detail in the manuscript text and explain why it is better suited than the ERGM framework in our case. The respective section can be found in the sections "Statistical analysis" and "Strength of the analytical approach".

Comment #2.9

2. I do not see a strong reason to capture diversity by categories such as "low", "medium", and "high"; the underlying measure is metric. Moreover, if categories must be used, I suggest using different thresholds for defining "low", "medium", and "high" diversity. I am not sure how classrooms with about .5 diversity can reasonably be described as "low" diversity settings.

Thank you for this suggestion. We firstly note that we agree that using diversity as a continuous variable retains more information in the analysis and is therefore generally beneficial to creating categories. For

this reason, in our main analysis (see Fig 3), we did not categorize the fractionalization index (i.e., the used measure of ethnic diversity) but treated it as a continuous variable instead.

At the same time, we had to categorize the fractionalization index to create Fig 2. This is because prior to the multivariate analysis, we aimed at showing empirically how the typical friendship and rejection networks of refugee students look under different diversity conditions and this required the creation of some categories. To create the three diversity categories, we simply took thirds of the analyzed classrooms with the lowest, medium, and the highest diversity in our sample to create the low, medium, and high categories, respectively. We are now describing this procedure briefly in the manuscript. However, we did not intend imply that "low", "medium" and "high" are absolute categories; instead, we simply aimed at showing that as diversity increases, the patterns of relationships change. We now emphasize this in the manuscript (see page 5).

We agree that even the lowest-diversity third of our sample is represented by diversity levels which may be considered rather high. However, we also note these levels may seem higher than they actually are. A value of .5, for instance, indicates that there is a 50% probability that two randomly selected peers from a school class have a different ethnic origin. This usually does not indicate a .5 proportion of minority-status individuals in a classroom because the fractionalization index also depends on how many different countries of origin are represented in the classroom. In our low-diversity classrooms, the average proportion of immigrant-origin students is 18% (S.D.=11%). In our medium-diversity classrooms, the average proportion is 39% (S.D.=13%). In our high-diversity classrooms, the average proportion is 69% (S.D.=16%). We now also include these numbers in our manuscript to better describe our diversity categories (see section "Measurement", page 10.)

To make sure that the general tendencies regarding diversity are robust to the thresholds chosen between diversity settings, we also conducted four robustness checks using alternative thresholds and diversity measures. These are described in the section Robustness checks in the main text and in SI Appendix B. The substantive results of these additional analyses displayed by Figures S2, S3, S4, and S5 support the main results presented in Fig 2.

Comment #2.10 (Clarity and context)

The text is mostly clear and accessible. However, the key term "social integration" is nowhere defined. Worse, the term seems to be used in two different meanings. On the one hand, in some parts of the manuscript, "social integration" seems to refer to social contacts between refugees and the native majority. For example, in the first paragraph of the introduction, the authors stress the importance of "peer relationships across ethnic boundaries because social integration is a key determinant of success in the school system and beyond." This is how the term is typically used in research on immigrants' integration (e.g., by Gordon or Alba), and, thus, how it is used in many of the references cited in the introduction (references 9-12). On the other hand, it later becomes clear that the authors seem to use the term "social integration" in reference to *any* kind of social contacts (e.g., in the third paragraph of the introduction or when referring to the number of friends of any group as an indicator of social integration). Given that the term is key for the manuscript, I recommend defining it early on and then using it in an unequivocally way.

Thank you for pointing out that our use of the term *social integration* was inconsistent in the manuscript. We define the term *social integration* in the new version of the manuscript. Specifically, the second and third paragraph of the new introduction sound as follows (see page 3):

"The social integration of adolescents refers to positive and supportive relationships, as indicated by friendships and a lack of peer rejection (Coleman 1990). Social integration improves adolescents' well-being (Motti-Stefanidi et al. 2020), health (Östberg and Modin 2008), and educational achievement (Stadtfeld et al. 2019). In contrast, low levels of social integration can have severe consequences for adolescents' psychological well-being (McDougall et al. 2001) and physical health (Wolke et al. 2013).

Friends are pivotal to social integration as they provide social capital, which includes resources such as valuable information and social support (Lin 2001; Portes 1998). However, the resources embedded in co-ethnic friendships often differ from those embedded in interethnic friendships (Lorenz, Boda, and Salikutluk 2021). In particular, immigrant students benefit from social contact with majority-group members (Lorenz, Boda, Salikutluk, et al. 2021). Such contacts provide access to resources such as exposure to the host country's language. Consequently, they enhance immigrant adolescents' opportunities to acquire critical resources such as language skills (Chiswick and Miller 2001) and, ultimately, success in the education system (Wölfer, Caro, and Hewstone 2019) and labor market (Kanas et al. 2012). Therefore, in addition to having positive peer relationships in general, establishing relationships across ethnic boundaries constitutes another key component of social integration for minority students (Nee and Alba 2013)."

Comment #2.11

Irrespective of the ambiguity of the term "social integration", the manuscript distinguishes between "friendship and rejection networks". "Rejection" is measured by the question "Who would you not want to sit next to?" in class. I understand that this probably measures a negative social relationship, but I am still puzzled whether it is accurate to say that a student "rejects" another one if she does not want to sit next to her in class. After all, there might be strategic reasons to avoid a specific peer as desk mate, for example, because they are too loud or maybe even because one is friends with her and she might thus be a distraction from studying. I am not saying that these examples are necessarily more accurate than the interpretation of the authors, but I do think that it is a bit of a stretch to conclude from this measure that "refugees are rejected". One solution might be to just always state "rejected as desk mates", another one would be to provide additional evidence that the item actually measures a genuine negative feeling towards another person.

Thank you for this comment. We agree that rejecting peers as a desk mate is not the same as not liking them. Therefore, following your suggestion, we use the term "rejection as desk mates" instead of just "rejection" whenever we discuss our results throughout the manuscript.

Reviewer #3:

Remarks to the Author:

Using a unique large nationally representative data set of social networks of refugee, non-refugee minority, and majority students, this study is the first to show the social integration of refugee students. This provides novel insights into how refugee students can be best accommodated. The study also shows the importance of the ethnic diversity of the local context (classroom) for students' willingness to socialize with newcomers.

In my view, this study is of immediate interest to policymakers and scholars studying social integration and social cohesion in many disciplines. The available high-quality data and the excellent application of sophisticated methods for social network analysis give credibility to the results. The analyses are adequately described in the supplemental material and a series of robustness checks increase trust in the conclusions drawn. The article and abstract are clearly written and should be accessible to a wide audience.

Thank you very much for the positive review and your helpful comments!

Comment #3.1

I have only a few notes on what the authors could do to strengthen the paper further:

I'm uncertain about the meaning of the finding that "first-generation immigrant students are more likely to both befriend and reject their refugee peers in more diverse classrooms" (p.7). From Figure 3, we learn that the probability of rejection in high diversity classrooms is similar to that of other groups. However, first-generation immigrant students are less likely to reject refugees in low-diversity classrooms. Importantly, there are relatively few first-generation students in the sample (472), and the logic of the analyses suggests that very few of them will be in low diversity classrooms. As a consequence, I wonder if the odd result of less rejection in low diversity classrooms is the consequence of a few extreme cases (outliers). An outlier analysis of the first-generation immigrant students would be reassuring that this finding is robust.

Thank you for this note on the low number of first-generation immigrants in low-diversity settings and the resulting uncertainty of the findings from these contexts. In response to your comment, we inspected those nominations descriptively. If the low-diversity results were driven by outliers, we could identify first-generation immigrants who do not reject (almost) any refugee classmates in low-diversity settings, and 2) this no-rejection behaviour should also be distinct from the typical nomination tendencies of other first-generation immigrants in the low-diversity settings. We can indeed identify many first-generation immigrant students who do not reject any of their refugee classmates – in fact, this is the case for the majority of first-generation students in these classrooms. Therefore, this cannot be considered outlier behaviour. Nevertheless, we agree that because there are only a few first-generation immigrants in our sample, and especially in classrooms at the low end of the diversity scale, our related results rely on small sample sizes. We now mention this in the discussion. On page 8, we write the following:

"However, we only have a relatively low number of first-generation immigrant students in our sample (N = 487), and by definition, only a small proportion of them attend classrooms with low diversity levels. Consequently, the findings related to diversity effects on first-generation immigrants' social-tie-creation behavior should be treated with caution."

Comment #3.2

What about the academic school level (tracking)? Schools of a higher academic track tend to be less ethnically diverse than those of a lower track. Is what we see, in part, related to academic tracking?

Thank you for this note on the possible role of tracking. To account for it, we conducted a new robustness check to see if students in different academic tracks befriend and reject their refugee peers differently. We find our main parameters of interest to be substantively similar in this model. For more details, see SI Appendix E and Table S10. In the section "Robustness checks" of the new manuscript, we write the following (see pages 12 and 13):

"Finally, we considered that students in different school tracks of the German education system might also be more or less likely to name refugee friends or reject them, independent of their own immigrant status. Therefore, additional models controlled for the attended school track and the interaction between the attended school track and the refugee status of the tie receiver (see SI Appendix E, Table S10). This is important because, in Germany, the secondary education system is organized into different school tracks, which differ in their socio-economic composition and the students' achievement and language skills."

Comment #3.3 (Minor notes)

The results in Figure 2 are very interesting but they might also lead to a biased interpretation of readers. One could read them to say that refugee students are mainly rejected by native students and – as soon as there are sufficient minority students – mainly befriended by minority students. However, this does not take the baseline probability into account. In low diversity classes, there are mainly native students who could send rejection ties. And in high diversity classes, there are relatively fewer native students who could send friendship ties. I would suggest adding descriptive analyses that take the baseline probability into account (e.g. proportions) – next to the analysis that are now shown.

Thank you for making this excellent point. The aim of our statistical analyses was, in fact, two-fold. First, we aimed to describe the kind of networks refugee adolescents are typically embedded in under different diversity conditions (descriptive analysis). This is the analysis presented in Fig 2. It does not control for the opportunity structure but the results show how the opportunity structure contributes to typical social networks of refugee students in different diversity contexts. Therefore, it leaves open whether the better social integration of refugees in more diverse classrooms is due to more contact with ethnic minority members (who might be more accepting towards refugees) or whether majority-group members have

different attitudes towards refugees under different diversity conditions. This is acknowledged on page 6 in the manuscript:

"Fig 2 shows students' social networks in classrooms with varying diversity. However, it does not provide any indications about the underlying processes. Are refugees in higher-diversity settings better socially integrated simply because of the presence of more ethnic minority students who may be more likely to accept them or because they develop more positive relations with peers from all ethnic groups, including native students?"

Disentangling these mechanisms was the aim of Fig 3. As we write on page 6, by controlling for the opportunity structure and several confounders, we can analyze the adolescents' preferences for accepting and rejecting refugee peers:

"To answer these questions, we apply linear regression models specifically developed to analyze social network data (Dekker, Krackhardt, and Snijders 2007). These models consider that observations in social networks (i.e., social ties) are not independent of each other."

The results show that it is not only non-refugee minority members' comparatively higher tendency to accept and not reject refugee peers that increases refugee students' social integration in more diverse schools but also the differential preferences among the majority group. Based on your comment and that of Reviewer #2, we highlight this again in the conclusion section of the new manuscript, where we write on page 7 the following:

"However, in more ethnically diverse classrooms, refugee adolescents are socially better integrated: they tend to have more friends and are rejected significantly less frequently as desk mates than in less diverse classrooms. This is revealed in descriptive analyses and multivariate social network models. The latter control for various factors relevant to social network dynamics, among them academic achievement and German language skills, which may make refugees less desirable as desk mates for their peers even in the absence of actual dislike. Notably, refugee students' improved social integration in more diverse classroom is not solely due to the preferences and higher shares of ethnic minority students, but variation in the preferences of majority-group adolescents for befriending and rejecting refugee peers across social contexts. In more diverse contexts, majority-group adolescents reject refugees as desk mates less often and tend to nominate them as friends more often than in less diverse contexts."

Comment #3.4

I would remove one "available" from this sentence (p. 4)

"Our analyses are based on the largest available dataset on refugee students' social networks currently available."

Thank you, we adapted the sentence accordingly.

Literature

Burnett Heyes, Stephanie, Yeou-Rong Jih, Per Block, Chii-Fen Hiu, Emily A. Holmes, and Jennifer YF Lau. 2015. "Relationship Reciprocation Modulates Resource Allocation in Adolescent Social Networks: Developmental Effects." *Child Development* 86(5):1489–1506.

Elmer, Timon, and Christoph Stadtfeld. 2020. "Depressive Symptoms Are Associated with Social Isolation in Face-to-Face Interaction Networks." *Scientific Reports* 10(1):1–12.

Snijders, Tom AB. "Statistical models for social networks." *Annual review of sociology* 37 (2011): 131-153.

Zajonc, R. B. Attitudinal effects of mere exposure. *Journal of Personality and Social Psychology* 9, 1–27 (1968).

Decision Letter, first revision:

10th October 2022

Dear Dr. Lorenz,

Thank you once again for your revised manuscript, entitled "Ethnic diversity fosters the social integration of refugee students," and for your patience during the re-review process.

Your manuscript has now been evaluated by two of the same reviewers who evaluated your original manuscript (Reviewers 2 and 3), as well as a new Reviewer 4 with expertise in Multiple Regression Quadratic Assignment Procedure (MRQAP) modelling. All reviewer feedback is included at the end of this letter. Although the reviewers found your manuscript to have improved during revision, they also raise some important outstanding concerns. We remain interested in the possibility of publishing your study in *Nature Human Behaviour*, but would like to consider your response to these remaining concerns in the form of a revised manuscript before we make a decision on publication.

You will see from their comments that Reviewers 2 and 3 are satisfied with the changes made in your revision. However, Reviewer 4 highlights several outstanding questions and concerns regarding the clarity and appropriateness of the methods. We ask that you carefully address each of this reviewer's points in your revision.

Additionally, please ensure that your revised manuscript continues to comply with our editorial policies and formatting requirements. Failure to do so will result in your manuscript being returned to you,

which will delay its consideration. To assist you in this process, I have attached a checklist that lists all of our requirements. If you have any questions about any of our policies or formatting, please don't hesitate to contact me.

In sum, we invite you to revise your manuscript taking into account all reviewer and editor comments. We are committed to providing a fair and constructive peer-review process. Do not hesitate to contact us if there are specific requests from the reviewers that you believe are technically impossible or unlikely to yield a meaningful outcome.

We hope to receive your revised manuscript within 4-8 weeks. I would be grateful if you could contact us as soon as possible if you foresee difficulties with meeting this target resubmission date.

- Include a "Response to the editors and reviewers" document detailing, point-by-point, how you addressed each editor and referee comment. If no action was taken to address a point, you must provide a compelling argument. This response will be used by the editors and reviewers to evaluate your revision.
- Highlight all changes made to your manuscript or provide us with a version that tracks changes.

[REDACTED]

We look forward to seeing the revised manuscript and thank you for the opportunity to review your work. Please do not hesitate to contact me if you have any questions or would like to discuss these revisions further.

Sincerely,
Aisha

Aisha Bradshaw, PhD
Senior Editor
Nature Human Behaviour

Reviewer expertise:

Reviewer #2: interethnic relations, student social networks, network analysis methods

Reviewer #3: interethnic relations, social networks

Reviewer #4: MRQAP methods, social networks

REVIEWER COMMENTS:

Reviewer #2:
Remarks to the Author:

Thank you very much for your thoughtful response to my comments. I am now convinced by the method; and I also appreciate the clarifications considering the terminology and possible policy implications.

Reviewer #3:
Remarks to the Author:

The authors have addressed all of my concerns appropriately. The revised version of the manuscript is excellent. I still believe that this is a very important study that is of great interest to both an academic and non-academic audience. Well done!

Signed: Tobias Stark

Reviewer #4:
Remarks to the Author:

This paper looks at the relationship between ethnic diversity in the classroom and migrant/refugee student rejection among 9th graders in Germany. This is certainly a worthwhile endeavor but there are a few issues in the paper that need addressing. First, there is not a clear distinction between various types of ethnic students that might be in the classroom as it relates to refugee/migration/asylum status. Not all migrants are refugees. The global networks associated with refugees, migrants, and asylum seekers are quite distinct in terms of network topology and other characteristics (e.g., legal status). Thus, the discussion needs to be clearer about the refugee/migrant/ asylum seeker similarities and differences and how this might ultimately influence study results. Starting on the bottom of page 4 the authors point to the possible important of such distinctions in the categorization of students. They provide a descriptive analysis comparing these student types. This descriptive comparison could benefit from even a simple statistical assessment across these distinctions (how statistically different are friend choice or rejection across the groups?). This goes for other descriptive comparisons (e.g., rejection rates described at the bottom of page 5). On page 5, second to the last paragraph, what the authors are describing is the network idea of homophily, or "birds of a feather flock together There are several social network measures of homophily, most notably the E-I Index,

that might be useful here” (although they do mention this in the course of operationalizing variables later in the manuscript). The data described is the key to this study and its potential contribution. The data collected looks to be quite impressive and certainly lends itself to a whole wide range of interesting research. The data appears to have been based on a random sample of schools involving 9th graders across three different secondary school types with a subsequent random sample of 9th grade classrooms in each of the randomly selected schools. Achievement data and other types of data, including social network data, were collected. If I understand the data correctly, students in the randomly selected classrooms reported on their various network relations with students in their classroom (as opposed to students in the school). This yielded 1,807 sociocentric or whole networks. Given the fact that refugees constitute only 2.2 percent of German population the analysis ultimately used 304 classrooms involving 6,390 students. The authors chose to use MR-QAP to model the data. Initially, I had difficulty understanding the choice of an MR-QAP approach in this case. If the authors are using this method of analysis there needs to be more information on how the data were constructed. First, QAP is a method for testing dyadic hypotheses. Normally, QAP and MR-QAP are used to analyze a single whole or sociocentric network (e.g., a company, global trade networks). In this case, a single classroom where every dyad has a true meaning (either i is friends with j or not). Thus, other types of dyadic relations (e.g., after school activities shared) can be compared to determine the presence or absence of friendship ties in the dependent friendship network. Since I assume the authors did not perform 304 individual MR-QAPs, I must conclude they included all 304 classrooms in a single run. There needs to be more information on how the matrices were constructed for use in the MR-QAP modeling. If all the 304 classrooms were included in a single matrix it would mean many dyads are impossible or undefined since there are no possible ties between members of different classrooms. This can be done, if the modeling only involves permutations within classrooms and ignores all undefined dyads. Undefined dyads would have to be excluded from the analysis. More information is needed on how the data were constructed for modeling.

Finally, in looking over the figures, particularly the series of ego networks in Figure 2, I’m curious why the authors did not use a simple personal network modeling approach using some form of GLMs. In this approach, the unit of analysis is the student rather than the dyad. Although it is true there is lack of independence within classrooms, it still seems to be a reasonable alternative way of modeling the data and certainly more in line with how survey data of this kind is typically analyzed in such cases. In sum, the data are interesting and the potential contribution of this work important, but there needs to be more clarification of how the data were organized for the analysis. And if MR-QAP was used were the permutations in the modeling only within meaningful dyads.

Author Rebuttal, first revision:**Reviewer #2:**

Thank you very much for your thoughtful response to my comments. I am now convinced by the method; and I also appreciate the clarifications considering the terminology and possible policy implications.

Thank you for your feedback and for helping us improve the manuscript.

Reviewer #3:

The authors have addressed all of my concerns appropriately. The revised version of the manuscript is excellent. I still believe that this is a very important study that is of great interest to both an academic and non-academic audience. Well done!

We are delighted about your positive feedback and thank you for helping us improve the manuscript.

Reviewer #4:

Comment #4.1

First, there is not a clear distinction between various types of ethnic students that might be in the classroom as it relates to refugee/migration/asylum status. Not all migrants are refugees. The global networks associated with refugees, migrants, and asylum seekers are quite distinct in terms of network topology and other characteristics (e.g., legal status). Thus, the discussion needs to be clearer about the refugee/migrant/ asylum seeker similarities and differences and how this might ultimately influence study results. Starting on the bottom of page 4 the authors point to the possible importance of such distinctions in the categorization of students. They provide a descriptive analysis comparing these student types. This descriptive comparison could benefit from even a simple statistical assessment across these distinctions (how statistically different are friend choice or rejection across the groups?).

Thank you for this note. We agree that the distinction between different forms of migration is relevant to our research question. We now address this issue by implementing several changes throughout the manuscript.

First, we would like to note that we differentiate between involuntary/forced immigrants (refugees) and other (first- and second-generation) immigrants in our analyses. Following your suggestions, we now include information on the significance of the group differences presented in Fig 1 (see page 5):

“The differences between the refugee and all non-refugee groups are statistically significant for both indicators of social integration.”

Second, we assume specific barriers to the social integration of refugee adolescents compared with other first- and second-generation-immigrant ethnic minority students. To illustrate some of these barriers empirically in our data, we added information on the significance of group differences in important background variables and our dependent variables, distinguishing between refugee, first-generation, second-generation, and native-majority students. Tables S3, S4, and S5 in SI Appendix B provide this information as part of our descriptive analyses. The results reveal that both native-majority and non-refugee ethnic minority adolescents differ from refugee adolescents regarding multiple attributes. These include, for instance, language skills and mathematics test scores. Note that our multivariate models control for all the background variables reported in Tables S3, S4, and S5 (see Table S2 in SI Appendix A).

Third, relating to the distinction between refugees and asylum seekers, we concede that there are differences between the three legal forms of protection granted to refugees in Germany (entitlement to asylum, refugee protection, and subsidiary protection), for instance, regarding the entitlement to privileged family reunification. However, we argue that these three groups have many key characteristics in common (e.g., push motivation, experiences of trauma, language barriers), which is why we subsumed the three

legal forms under the label “refugee”. As a result of these considerations and your comments, we now consistently use the term *refugee* throughout the manuscript. The new manuscript now additionally includes the following sentence on page 10:

„These students typically receive one of three forms of legal protection granted to forced migrants in Germany: entitlement to asylum, refugee protection, or subsidiary protection.“

Comment #4.2

This goes for other descriptive comparisons (e.g., rejection rates described at the bottom of page 5).

We agree that including tests of statistical significance is helpful here as well. In terms of the friendship differences among refugee adolescents between the diversity settings (page 5), we now note the following:

“The difference is only significant between the low- and high-diversity settings.”

Regarding rejection rates (page 6), we now also add similar information. The manuscript now reads:

“Overall, refugee students are rejected as desk mates by 42% fewer classmates in high-diversity settings than in low-diversity settings, with the difference in rejection rates being significant between each pair of diversity settings.”

Comment #4.3

On page 5, second to the last paragraph, what the authors are describing is the network idea of homophily, or “birds of a feather flock together. There are several social network measures of homophily, most notably the E-I Index, that might be useful here” (although they do mention this in the course of operationalizing variables later in the manuscript).

We agree that the tendency among refugee adolescents to befriend refugee peers reflects a strong tendency for homophily. We highlight this in the new manuscript on page 5, where we write the following:

“Importantly, Fig 2 also shows that a refugee student has, on average, one refugee friend across all diversity settings. This holds even though the majority of the classrooms include only one or two refugee students, with a mean of 1.6 refugee students per class in the analyzed sample. Thus, refugee students appear to be very likely to befriend each other if more than one of them is present in a classroom. This finding reflects the well-established phenomenon of homophily (McPherson et al. 2001), which describes that people tend to build social ties with those who are similar in terms of salient attributes, such as ethnic origin or flight experience.”

In our MRQAP results, homophily among refugee adolescents is reflected in the positive and significant effects of the dyadic variable “refugee → refugee”, but also in several other dyadic variables included in our models, such as “same gender” or “similarity in language skills” (see Table S2 in SI Appendix A).

Comment #4.4

The authors chose to use MR-QAP to model the data. Initially, I had difficulty understanding the choice of an MR-QAP approach in this case. If the authors are using this method of analysis there needs to be more information on how the data were constructed. First, QAP is a method for testing dyadic hypotheses. Normally, QAP and MR-QAP are used to analyze a single whole or sociocentric network (e.g., a company, global trade networks). In this case, a single classroom where every dyad has a true meaning (either i is friends with j or not). Thus, other types of dyadic relations (e.g., after school activities shared) can be compared to determine the presence or absence of friendship ties in the dependent friendship network. Since I assume the authors did not perform 304 individual MR-QAPs, I must conclude they included all 304 classrooms in a single run. There needs to be more information on how the matrices were constructed for use in the MR-QAP modeling. If all the 304 classrooms were included in a single matrix it would mean many dyads are impossible or undefined since there are no possible ties between members of different classrooms. This can be done, if the modeling only involves permutations within classrooms and ignores all undefined dyads. Undefined dyads would have to be excluded from the analysis. More information is needed on how the data were constructed for modeling.

Thank you very much for pointing this out. We agree that the way we combined our various classroom-level matrices in one analysis is a very important issue, and we missed an opportunity to discuss this in our manuscript. Our procedure is in line with your suggestion (and with other published articles using multigroup MR-QAP models, e.g., Burnett Heyes et al, 2015; Elmer and Stadtfeld, 2020). That is, the permutations were restricted to within-classroom dyads and ignored undefined (between-classroom) dyads. We now include a sentence about this in the manuscript, page 11:

“Multigroup MRQAPs restrict permutations to within-classroom dyads and ignore the substantively meaningless between-classroom dyads (Elmer and Stadtfeld, 2020).”

Comment #4.5

Finally, in looking over the figures, particularly the series of ego networks in Figure 2, I'm curious why the authors did not use a simple personal network modeling approach using some form of GLMs. In this approach, the unit of analysis is the student rather than the dyad. Although it is true there is lack of independence within classrooms, it still seems to be a reasonable alternative way of modeling the data and certainly more in line with how survey data of this kind is typically analyzed in such cases.

We initially thought the same way and started with an ego-networks approach using GLMs. The reason we decided to combine a descriptive ego-network analysis (Fig 2) with a whole-network modeling approach (Fig 3) was to be able to account for baseline diversity-related differences between classrooms and disentangle them from diversity effects specific to refugee students. Specifically, in our sample, all students received fewer rejection nominations on average in more diverse classrooms, not only refugees. On the one hand, we acknowledge that even in this case, refugees are still better integrated in more diverse settings: this is why we still include the ego networks in Figure 2, which show the "gross" level of integration of refugees in different diversity levels (without any control variables, or accounting for baseline diversity differences). On the other hand, whether the effect of diversity is (partly) refugee-specific also matters, especially theoretically. Because of this, we deemed it necessary to also distinguish between the baseline diversity effect and the specific effect of diversity on the social integration of refugees. For this reason, we opted for an additional whole-network approach (Fig 3). We have now included a few words about the choice of our analytical strategy in the new version of the manuscript (page 12):

"We combine an ego-network approach for descriptive analysis with a whole-network approach for multivariate statistical modeling. We first show the overall social integration of refugees under different diversity conditions by presenting refugee ego networks (Fig 2). Then, we employ multivariate statistical modeling to explain the social mechanisms that produce these ego networks while taking into various characteristics of refugees that typically play a role in social-tie choices (e.g., age, language skills, academic achievement, etc.; Fig 3). The whole-network approach allows us to control for peer characteristics, the role of network processes (e.g., reciprocity, transitivity, etc.), and baseline differences in the social integration of adolescents based on different diversity levels. In this way, we can account for whether classrooms with higher levels of diversity may provide all students with higher levels of social integration, not only refugee students. Combining a descriptive ego-network approach with whole-network-based statistical modeling enables us to show the overall social integration of refugee students in different diversity settings and provide a thorough insight into the social mechanisms behind such diversity-based differences."

References

Burnett Heyes, Stephanie, Yeou-Rong Jih, Per Block, Chii-Fen Hiu, Emily A. Holmes, and Jennifer YF Lau. 2015. "Relationship Reciprocation Modulates Resource Allocation in Adolescent Social Networks: Developmental Effects." *Child Development* 86(5):1489–1506.

Elmer, Timon, and Christoph Stadtfeld. 2020. "Depressive Symptoms Are Associated with Social Isolation in Face-to-Face Interaction Networks." *Scientific Reports* 10(1):1–12.

Decision Letter, second revision:

5th January 2023

Dear Dr. Lorenz,

Thank you for submitting your revised manuscript "Ethnic diversity fosters the social integration of refugee students" (NATHUMBEHAV-211117168B). It has now been seen by one of the original referees, and their comments are below. As you can see, the reviewer finds that the paper has improved in revision. We will therefore be happy in principle to publish it in *Nature Human Behaviour*, pending minor revisions to satisfy the referees' final requests and to comply with our editorial and formatting guidelines.

We are now performing detailed checks on your paper and will send you a checklist detailing our editorial and formatting requirements within approximately two weeks. Please do not upload the final materials and make any revisions until you receive this additional information from us.

Sincerely,
Aisha

Aisha Bradshaw, PhD
Senior Editor
Nature Human Behaviour

Reviewer #4 (Remarks to the Author):

The authors have addressed most of my concerns. I suggest when reporting on the statistical significance of any analysis that they include the significance and test statistic results at the end of the sentence in parentheses. Example: the first paragraph end of last sentence on page 6 "with the difference in rejection rates being significant between each pair of diversity settings ($p < 0.xx$, test statistic value). Same would be true for the highlighted sentences on page 5. Just a suggestion on the theoretical and historical background. There were some early studies that used social networks to examine the relationship between networks and rejection in classrooms and the psychological ramifications of such rejections. In addition, the work used a QAP approach similar to this manuscript. Here are the references.

J.C. Johnson, M. Ironsmith, A.L. Whitcher, G.M. Poteat, and C.W. Snow. "The Development of Social Networks in Preschool Children." *Early Education and Development* 8(4) 389-406, 1997.

J.C. Johnson, M. Ironsmith, and G. Michael Poteat. "Assessing Children's Sociometric Status: Issues and the Application of Social Network Analysis," *Journal of Group Psychotherapy, Psychodrama, and Sociometry* 47(1):36-48, 1994.

J.C. Johnson, G.M. Poteat, and M. Ironsmith. "Structural Vs. Marginal Effects: A Note on the Importance of Structure in Determining Sociometric Status," *Journal of Social Behavior and Personality* 6(3):489-508, 1991.

Author Rebuttal, second revision:

Reviewer #4:

Comment #4.1

The authors have addressed most of my concerns. I suggest when reporting on the statistical significance of any analysis that they include the significance and test statistic results at the end of the sentence in parentheses. Example: the first paragraph end of last sentence on page 6 "with the difference in rejection rates being significant between each pair of diversity settings ($p < 0.xx$, test statistic value). Same would be true for the highlighted sentences on page 5.

Thank you for your positive feedback and this suggestion. We added the p-values and the test statistic to the text wherever this was applicable. For MR-QAP analyses, statistical significance was tested directly using a permutation test: each p-value equals the proportion of simulated cases in which we see a parameter at least as large (in absolute value) as the parameter for the observed network (for a detailed explanation, see Supplementary Appendix F). Therefore, in cases, where we report the statistical significance of MR-QAP results, we only present the p-value in the text. The changes in the manuscript are highlighted in yellow.

Comment #4.2

Just a suggestion on the theoretical and historical background. There were some early studies that used social networks to examine the relationship between networks and rejection in classrooms and the psychological ramifications of such rejections. In addition, the work used a QAP approach similar to this manuscript. Here are the references.

J.C. Johnson, M. Ironsmith, A.L. Whitcher, G.M. Poteat, and C.W. Snow. "The Development of Social Networks in Preschool Children." *Early Education and Development* 8(4) 389-406, 1997.

J.C. Johnson, M. Ironsmith, and G. Michael Poteat. "Assessing Children's Sociometric Status: Issues and the Application of Social Network Analysis," *Journal of Group Psychotherapy, Psychodrama, and Sociometry* 47(1):36-48, 1994.

J.C. Johnson, G.M. Poteat, and M. Ironsmith. "Structural Vs. Marginal Effects: A Note on the Importance of Structure in Determining Sociometric Status," *Journal of Social Behavior and Personality* 6(3):489-508, 1991.

Thank you for the suggestion. We now mention the 1991 paper in the discussion section, where we describe that social integration refers, besides positive (friendship) nomination, also to the lack of negative nominations and peer rejection (see page 3). We additionally rely on the 1997 paper in the section describing that standard statistical methods cannot be used to investigate social network processes (see page 11).

Final Decision Letter:

Dear Dr. Lorenz,

We are pleased to inform you that your Article "Ethnic diversity fosters the social integration of refugee students", has now been accepted for publication in *Nature Human Behaviour*.

Please note that *Nature Human Behaviour* is a Transformative Journal (TJ). Authors whose manuscript was submitted on or after January 1st, 2021, may publish their research with us through the traditional subscription access route or make their paper immediately open access through payment of an article-processing charge (APC). Authors will not be required to make a final decision about access to their article until it has been accepted. IMPORTANT NOTE: Articles submitted before January 1st, 2021, are not eligible for Open Access publication. Find out more about Transformative Journals

We welcome the submission of potential cover material (including a short caption of around 40 words) related to your manuscript; suggestions should be sent to Nature Human Behaviour as electronic files (the image should be 300 dpi at 210 x 297 mm in either TIFF or JPEG format). Please note that such pictures should be selected more for their aesthetic appeal than for their scientific content, and that colour images work better than black and white or grayscale images. Please do not try to design a cover

with the Nature Human Behaviour logo etc., and please do not submit composites of images related to your work. I am sure you will understand that we cannot make any promise as to whether any of your suggestions might be selected for the cover of the journal.

With best regards,
Aisha

Aisha Bradshaw, PhD
Senior Editor
Nature Human Behaviour